# Platelet-derived chemokines promote skeletal muscle regeneration by guiding neutrophil recruitment to injured muscles

Flavia A. Graca[1], Anna Stephan[1], Benjamin A. Minden-Birkenmaier[1,2], Abbas Shirinifard[1], Yong-Dong Wang [3], Fabio Demontis [1,4] ✉ & Myriam Labelle [1,2,4] ✉

Skeletal muscle regeneration involves coordinated interactions between different cell types. Injection of platelet-rich plasma is circumstantially considered an aid to muscle repair but whether platelets promote regeneration beyond their role in hemostasis remains unexplored. Here, we find that signaling via platelet-released chemokines is an early event necessary for muscle repair in mice. Platelet depletion reduces the levels of the platelet-secreted neutrophil chemoattractants CXCL5 and CXCL7/PPBP. Consequently, early-phase neutrophil infiltration to injured muscles is impaired whereas later inflammation is exacerbated. Consistent with this model, neutrophil infiltration to injured muscles is compromised in male mice with Cxcl7-knockout platelets. Moreover, neo-angiogenesis and the re-establishment of myofiber size and muscle strength occurs optimally in control mice post-injury but not in Cxcl7ko mice and in neutrophil-depleted mice. Altogether, these findings indicate that platelet-secreted CXCL7 promotes regeneration by recruiting neutrophils to injured muscles, and that this signaling axis could be utilized therapeutically to boost muscle regeneration.

Skeletal muscle has the remarkable capacity to repair injuries that occur in response to trauma, xenobiotics, and strenuous exercise[1-5]. In addition to physiological injuries, muscle damage and regeneration occurs in muscular dystrophies[4,6,7] and cancer-induced cachexia[8], and it is defective in aging[9,10].

Previous studies have found that muscle regeneration starts immediately after the death of myofibers, which leads to the recruitment of immune cells to the muscle. These infiltrating cells are necessary for muscle regeneration via their capacity to remove cellular debris and promote myogenesis, i.e., the fusion of satellite muscle stem cells to form new myofibers[2,3,11-13]. Several waves of immune cell recruitment occur during muscle regeneration, and these include neutrophils, monocytes, macrophages, and T cells[14]. Neutrophils are recruited in the early phase of muscle regeneration and are followed by M1/M2 macrophages in later phases[2,3,11-13]. Whereas neutrophils and M1 macrophages contribute to inflammation, M2 macrophages have anti-inflammatory functions[2,3,11-13,15]. Partial depletion of neutrophils and macrophages impairs muscle regeneration[16-19], indicating that invading immune cells are indeed necessary for muscle repair. Importantly, optimal muscle regeneration requires robust but transient recruitment of distinct immune populations, and the transition from an inflammatory to an anti-inflammatory state[2,3,11-13]. Consequently, persistent inflammation is a cause of myopathies and age-related muscle repair deficits[20-24].

While macrophages have been extensively investigated[19,25,26], relatively less is known about the role of neutrophils in tissue repair[27-30]

[1]Department of Developmental Neurobiology, St. Jude Children's Research Hospital, Memphis, TN 38105, USA. [2]Department of Oncology, Division of Molecular Oncology, St. Jude Children's Research Hospital, Memphis, TN 38105, USA. [3]Department of Cell and Molecular Biology, St. Jude Children's Research Hospital, Memphis, TN 38105, USA. [4]These authors contributed equally: Fabio Demontis, Myriam Labelle. ✉e-mail: Fabio.Demontis@stjude.org; Myriam.Labelle@stjude.org

and muscle regeneration[2,3,11–13,18]. Upon injury, neutrophils rapidly invade the damaged muscle, where they remove cellular debris and secrete inflammatory factors that recruit monocytes and macrophages[2,3,11–13,18]. Eventually, neutrophils move back to the circulation starting from 24 h after injury[2,3,11–13]. Altogether, neutrophils are pivotal for the phagocytosis of necrotic material and for stimulating the homing of other inflammatory cells[2,3,11–13,18]. However, persistent infiltration of neutrophils into muscles exacerbates inflammation, causes further damage, and delays the subsequent steps of muscle regeneration[6,31–33]. Consequently, it was found that depleting neutrophils starting from 24 h after injury accelerates the transition to the subsequent phases of muscle regeneration and hence improves muscle repair[34]. Therefore, a rapid and robust but transient recruitment of neutrophils is necessary for efficient muscle regeneration. Despite their importance, the mechanisms responsible for the timely recruitment of neutrophils to injured muscles are incompletely understood. Previous studies have found that necrotic myofibers activate the complement system and muscle-resident mast cells and neutrophils, which in turn release pro-inflammatory factors that promote the massive recruitment of additional neutrophils from the circulation[11,32,33,35,36]. However, it remains undetermined whether neutrophil recruitment from the bloodstream is simply dependent on cues released from injured myofibers and muscle-resident cells or whether other signals are also necessary.

Platelets are anucleated blood cells derived from megakaryocytes and known for their role in hemostasis[37–40]. Specifically, platelets are activated by many stimuli associated with blood vessel injury and are responsible for clot formation in the immediate phase that follows injury[41,42]. Upon activation, platelets release the contents of alpha and dense secretory granules, which include cytokines, growth factors, and metabolites[43–46]. Platelet-secreted factors have been found to contribute to many processes beyond coagulation[47–49]. Interestingly, several studies have explored the therapeutic use of the platelet-rich plasma (PRP), i.e., the component of the blood devoid of white and red blood cells but rich in platelets and platelet-secreted factors (releasate). In particular, injection of the PRP has been proposed to boost wound healing and regeneration in a number of tissues[50–58], including skeletal muscle[59–64]. However, it remains largely undetermined whether platelets are necessary for skeletal muscle regeneration beyond their role in hemostasis, and whether platelet-secreted factors are required for any step of muscle repair.

Here, we show that platelet-secreted chemokines guide the early steps of muscle repair by recruiting neutrophils to injured muscles. Perturbation of this early step of regeneration compromises the re-establishment of myofiber size and muscle strength in mice. Altogether, these findings indicate a key role for platelet-derived signals in initiating skeletal muscle regeneration.

## Results
### Platelets are detected in skeletal muscles upon injury and this can be prevented via antibody-based platelet depletion
To start to investigate whether platelets contribute to skeletal muscle regeneration, we have first examined via immunostaining whether platelets are present in skeletal muscles upon injury. To this purpose, we have utilized an experimental model of injury induced by intramuscular injection of cardiotoxin (CTX) into the tibialis anterior (TA) skeletal muscle of mice[65–67], (Fig. 1a). Whereas no platelets are found in uninjured muscles, platelet aggregates were prominent at day 1, were present at day 7, and eventually declined at day 14 after CTX injection (Fig. 1b–d). Co-immunostaining with the endothelial cell marker PECAM-1 indicated that platelet aggregates, which have pro-hemostatic effects[68–70], are located primarily within blood vessels of injured skeletal muscles (Supplementary Fig. 1).

To determine the functional significance of platelet recruitment to injured muscles, we next examined skeletal muscle regeneration in the absence of platelets. To this purpose, 2 h before intramuscular CTX injection, mice were injected via the tail vein with a platelet-depleting antibody, which has been previously shown to reliably ablate platelets within 30 min from injection and for up to 3-4 days from injection[71–74]. Additionally, a second dose of platelet-depleting antibody was injected at day 4 following CTX injection (Fig. 1a). Control mice were treated in the same way but injected with a mock IgG antibody, as previously done in other studies that utilized these tools[71–74] (Fig. 1a). As expected, tail-vein injection of the platelet-depleting antibody resulted in the consequent lack of platelets in injured TA muscles at day 1 and 7 after CTX injection (Fig. 1b–d). Altogether, this indicates that platelets localize to skeletal muscle early after injury, and that this is prevented by antibody-based platelet depletion.

### Platelets are necessary for neutrophil recruitment to injured skeletal muscles
Muscle regeneration relies on the recruitment of immune cells (e.g., neutrophils and macrophages) that are tasked with the removal of cellular debris and the subsequent promotion of myogenesis[2–4,11]. Neutrophils localize to injured muscles in the early phase of regeneration and set the stage for the subsequent invasion of regenerating muscles by macrophages[12,13,31]. However, the mechanisms responsible for the recruitment of neutrophils from the bloodstream are incompletely understood.

Because platelet aggregates are found in skeletal muscles early after injury (Fig. 1a–d), we hypothesize that they may regulate the recruitment of neutrophils to injured muscles. To test this model, hematoxylin/eosin (H&E) staining of TA muscle sections was utilized to determine the impact of antibody-mediated platelet depletion on muscle regeneration. Whereas infiltrates of immune cells were clearly seen at day 1 after injury in mock-treated mice, such invading immune cells were nearly absent in TA muscles obtained from platelet-depleted mice (Fig. 2a, b). To further test these findings, muscle sections were stained with known neutrophil markers, MMP-9 and Ly6G. As expected based on previous studies[2,3,11–13], neutrophils were rare in uninjured muscles but abundant at day 1 after CTX injection. Platelet depletion significantly reduced neutrophil recruitment to muscles at day 1 after CTX injection whereas no substantial effects were seen at day 7 and 14 after injury (Fig. 2a–e), timepoints at which there is minimal presence of neutrophils in muscle[2,3,11–13]. Altogether, these findings indicate that platelet depletion impairs neutrophil recruitment to injured skeletal muscles.

Normally, staggered waves of immune cells are recruited to skeletal muscle upon injury to remove cell debris, to promote the recruitment of subsequent populations of immune cells, and to promote myogenesis[2,3,11–13]. On this basis, we next examined the impact of platelet depletion on the infiltration of macrophages, which are recruited to injured muscles in later phases of muscle regeneration. As expected[2,3,11–13], immunostaining of muscle sections revealed that macrophages are more abundant in skeletal muscles at day 7 compared to uninjured muscles and to day 1 after injury (Fig. 2f and Supplementary Fig. 2). Interestingly, platelet depletion significantly increased overall macrophage infiltration into regenerating muscles at day 7 from injury (Fig. 2f and Supplementary Fig. 2), and this was due to higher levels of both M1 and M2 macrophages (Supplementary Fig. 3). These findings might be explained by the fact that reduced neutrophil recruitment due to platelet depletion may impede the removal of cell debris in the early phases of regeneration, which may exacerbate inflammation and lead to a higher macrophage recruitment at later stages.

### Platelets are necessary for the growth of newly-formed myofibers and for the optimal regeneration of injured muscles
Optimal muscle regeneration depends on waves of immune cell recruitment to injured muscles and their interaction with regenerating

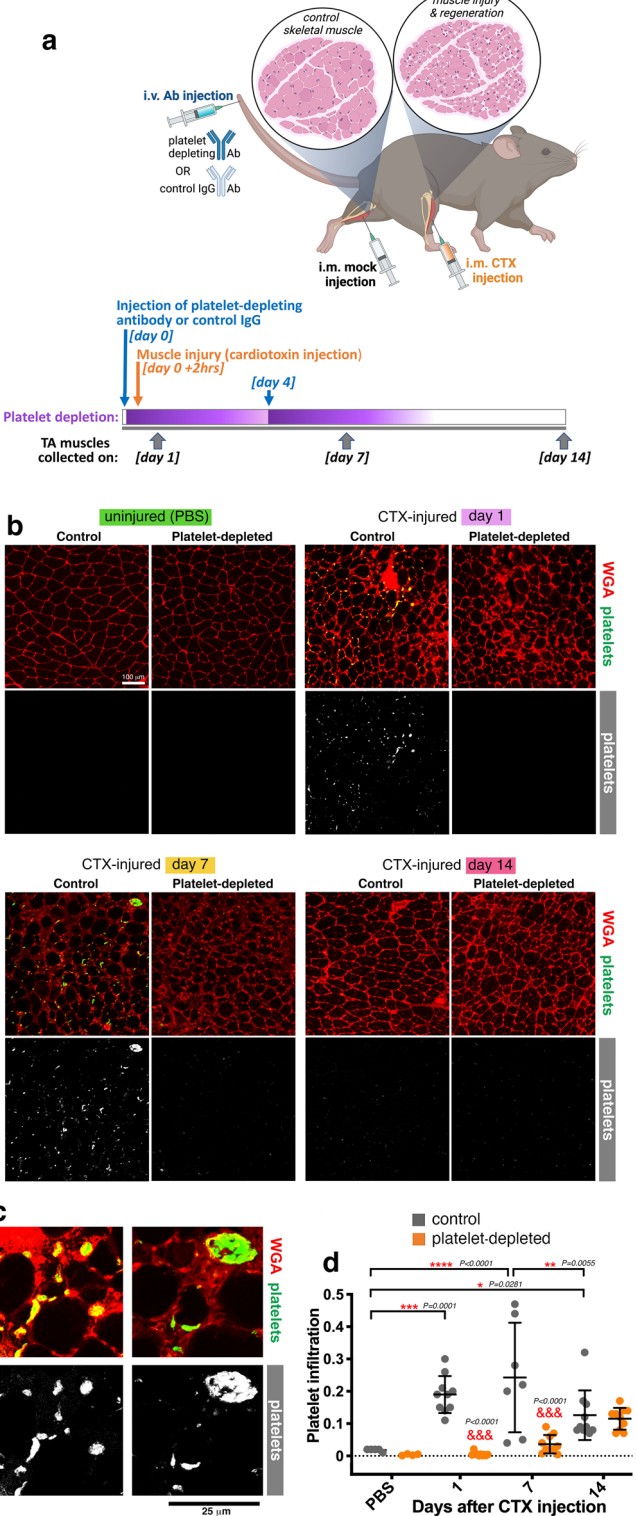

**Fig. 1 | Platelet thrombi are found in injured muscles and are prevented via antibody-based platelet depletion. a** Experimental strategy to assess the role of platelets in skeletal muscle regeneration. The i.v. injection of a platelet-depleting antibody is done 2 h before muscle injury and is repeated at day 4 after injury. Control mice are injected with an IgG control antibody. Muscle injury is induced via the injection of cardiotoxin (CTX) into a tibialis anterior (TA) muscle whereas the contralateral TA muscle is mock-injected with PBS. TA muscles are retrieved at day 1, 7, and 14 for further analyses. **b–d** Immunostaining of TA muscles from control and platelet-depleted mice, either injured via the injection of cardiotoxin (CTX) or uninjured (mock-injected with PBS). WGA (red) provides an outline of myofibers whereas platelet aggregates (green) of different sizes are detected with anti-GP1bβ antibodies. Platelet thrombi are found in injured muscles at day 1 and day 7 from CTX-induced injury but their presence is minimal at day 14 from injury and they are not detected in uninjured TA muscles. Antibody-based platelet depletion results in the lack of platelets aggregates in injured muscles, indicating that this strategy is effective for testing the role of platelets in muscle regeneration. The graph displays the mean ±SD with $n = 5$ (from 5 control independent mice) and $n = 4$ (from 4 platelet-depleted independent mice) biologically independent uninjured muscles; $n = 7$ biologically independent CTX-injured muscles obtained from $n = 7$ independent control mice at day 7 after injury; and $n = 10$ biologically independent CTX-injured muscles obtained from $n = 10$ independent mice for each of the other timepoints and conditions analyzed; *$P < 0.05$, **$P < 0.01$, ***$P < 0.001$, ****$P < 0.0001$ (two-way ANOVA with Tukey post hoc test); &&&$P < 0.001$ (two-way ANOVA with Sidak post hoc test) refers to the comparison of muscles from control versus platelet-depleted mice at a given timepoint of regeneration. Source data are provided in the Source data file.

minimal diameter indicates that platelet depletion reduces the size of myofibers at day 14 but not at day 7 post-injury. Collective analysis of all myofibers derived from the TA muscles of each group further indicates that platelet depletion overall decreases the range of myofiber sizes at day 14 post-injury (Fig. 3d).

The decline in myofiber size due to platelet depletion may depend on an impediment of myogenesis, i.e., the formation of new myofibers from satellite muscle stem cells. To examine whether myogenesis is impacted by platelet depletion, we measured the abundance of eMHC-positive myofibers. Normally, eMHC (embryonic myosin heavy chain) is expressed during muscle development and disappears after birth but it is re-expressed during myogenesis associated with muscle regeneration[75–77]. In agreement with this knowledge, there are many eMHC-positive myofibers at 7 days after cardiotoxin-induced injury, coincident with myogenesis, but their abundance declines at day 14 post-injury (Supplementary Fig. 4), a timepoint at which myogenesis and muscle regeneration are largely resolved. Similar percentages of eMHC-positive tissue areas are found in regenerating skeletal muscles from control and platelet-depleted mice (Supplementary Fig. 4), suggesting that platelet depletion does not impair myogenesis. However, our finding that myofiber size is reduced 14 days post-injury in platelet-depleted mice (Fig. 3) suggests that the lack of platelets impairs the growth of newly-formed myofibers.

## Cytokine profiling of regenerating skeletal muscles indicates that platelet depletion impairs neutrophil chemotactic signaling, neo-angiogenesis, and myofiber growth

Upon activation, platelets release the content of secretory granules, which consists of cytokines, growth factors, and metabolites[43–46]. Previous studies have found that platelet-secreted factors can contribute to tissue repair[47–49]. On this basis, platelet depletion may impact muscle regeneration because of decreased levels of platelet-secreted factors and/or modulation of cytokine production by muscle cells and infiltrating immune cells.

To test this hypothesis, we have utilized Quantibody (Quantitative Multiplex ELISA) arrays to profile the protein levels of 640 mouse cytokines in extracts from TA skeletal muscles obtained from

myofibers[2,3,11–13]. We have found that platelet depletion impairs the timely recruitment of neutrophils to injured muscles (Fig. 2).

On this basis, we next examined muscles at 1, 7, and 14 days after CTX injection to test whether there are consequent effects on skeletal muscle regeneration. Staining with phalloidin and anti-Laminin antibodies revealed that muscle regeneration is impeded by platelet depletion, as indicated by the lower size of myofibers of TA muscles from platelet-depleted mice at day 14 post-injury (Fig. 3a–c). Specifically, measurement of the myofiber size via estimation of the Feret's

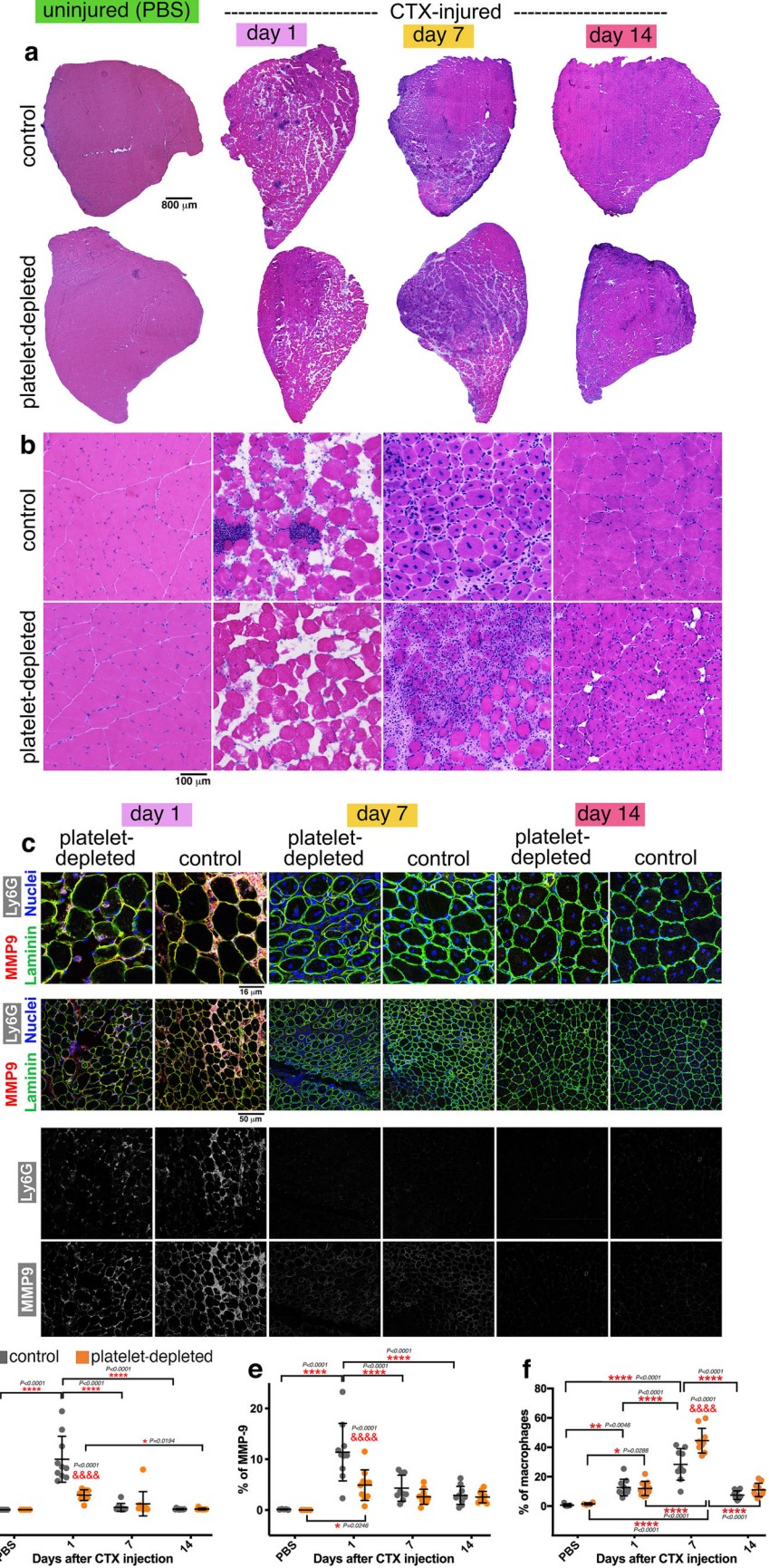

**Fig. 2 | Neutrophil infiltration in injured muscles is impeded by platelet depletion whereas late-phase macrophage infiltration is increased. a, b** H&E staining of TA muscles from control and platelet-depleted mice at day 1, 7, and 14 from cardiotoxin (CTX)-induced injury. In agreement with previous studies, immune infiltration is found at day 1 after CTX in control TA muscles but it is largely reduced in the TA muscles from platelet-depleted mice. At later stages, the process of muscle regeneration is impaired, as indicated by the overall lower size of myofibers and by the ultrastructural defects of TA muscles that are found at day 14. There are no noticeable changes in uninjured muscles from platelet-depleted versus control mice. **c** Immunostaining of TA muscles from control and platelet-depleted mice for neutrophil markers, i.e., MMP9 (red) and Ly6G (white). Myofiber boundaries are identified with immunostaining for anti-Laminin antibodies (green) whereas nuclei are identified by DAPI (blue). **d**–**e** Neutrophil infiltration in injured muscles occurs predominantly at day 1 from injury, and it is significantly reduced by platelet depletion. Similar results are found with the quantitation of both Ly6G and MMP9 immunostaining. **f** Quantitation of macrophages infiltrating the muscle, as defined with anti-F4/80 antibodies. Macrophage infiltration is predominant at day 7 from CTX-mediated injury and it is exacerbated by platelet depletion. In d-f, the graphs display the mean ±SD with $n = 5$ (from 5 control independent mice) and $n = 4$ (from 4 platelet-depleted independent mice) biologically independent uninjured muscles; $n = 9$ biologically independent CTX-injured muscles obtained from $n = 9$ independent control mice at day 7 after injury; and $n = 10$ biologically independent CTX-injured muscles obtained from $n = 10$ independent mice for each of the other timepoints and conditions analyzed; *$P < 0.05$, **$P < 0.01$, ***$P < 0.001$, ****$P < 0.0001$ (two-way ANOVA with Tukey post hoc test); &&&&$P < 0.0001$ (two-way ANOVA with Sidak post hoc test) refers to the comparison of muscles from control versus platelet-depleted mice at a given timepoint of regeneration. Source data are provided in the Source data file.

mice with or without antibody-mediated platelet depletion at 1, 7, and 14 days after CTX-induced muscle injury. In addition, we also profiled the levels of cytokines in uninjured muscles 7 days after platelet depletion.

As expected, the levels of several cytokines were modulated at different timepoints of muscle regeneration but the highest difference in the profile of secreted factors between platelet-depleted versus controls was found at day 1 after regeneration (Fig. 4a, b). GO term analysis of significantly-regulated cytokines in platelet-depleted versus control TA muscles indicates that several categories of cytokines are collectively regulated at different timepoints of muscle regeneration in a platelet-dependent manner (Fig. 4c). At day 1 after injury, platelet depletion leads to a remarkable decline in the levels of cytokines that promote neutrophil chemotaxis. At this early timepoint, there is also a significant decline in the levels of VEGF (Supplementary Fig. 6a), a potent inducer of blood vessel formation which can be secreted by platelets as well as by neutrophils during tissue regeneration[30,78–82]. Coincident with the reduction in VEGF levels, we find that post-injury muscles from platelet-depleted mice have defects in neo-angiogenesis, as indicated by immunostaining for PECAM1 (Supplementary Fig. 6b, c).

In later phases of muscle regeneration (day 7 and 14), platelet depletion resulted in increased levels of cytokines associated with the inflammatory response (Fig. 4c), which is suggestive of excessive and persistent inflammation that may impair regeneration and myofiber growth[83]. In particular, tumor necrosis factor ligands (TNFs) were significantly upregulated in muscles from platelet-depleted mice at day 7 (Supplementary Fig. 5). Although transient TNF signaling promotes myogenesis during muscle regeneration[84–87], activation of this pathway has been found to stunt myofiber growth and to induce myofiber atrophy[88–91]. Therefore, the upregulation of TNF ligands in the regenerating muscles of platelet-depleted mice likely contributes to the reduced myofiber size that is found in these muscles post-injury (Fig. 3).

We have found that platelet depletion reduces the infiltration of neutrophils in regenerating skeletal muscles. On this basis, the lack of platelets may decrease neutrophil recruitment to injured muscles because of decreased levels of platelet-secreted cytokines necessary for neutrophil chemotaxis. Consistent with this hypothesis, the levels of several chemokines reported to promote neutrophil chemotaxis[92,93] were reduced (Fig. 4d). Specifically, the protein levels of CXCL1, CXCL2, CXCL4/PF4, CXCL5, and CXCL7/PPBP increase at day 1 after injury, coincident with the phase of neutrophil recruitment to injured skeletal muscle. However, such increase in the protein levels of chemokines that promote neutrophil recruitment is impeded by platelet depletion. Quantitatively, CXCL4/PF4 and CXCL7/PPBP have concentrations ~1000x higher than CXCL1, CXCL2, and CXCL5 (Fig. 4d), suggesting that these chemokines may have a predominant role.

CXCL4/PF4, CXCL5, and CXCL7/PPBP are CXC chemokines that are nearly exclusively expressed in megakaryocytes and platelets[94–98].

Beyond this cell type-specificity of expression, platelets are the most abundant source of bioactive CXCL4/PF4, CXCL5, and CXCL7/PPBP because these chemokines are stored in α-granules and can be immediately released at high (µM) concentrations upon platelet activation[94–98].

Although these chemokines have been previously reported to recruit neutrophils, in vitro migration assays with recombinant versions of these chemokines indicate that CXCL5 and CXCL7 have the strongest chemotactic functions whereas recombinant CXCL4/PF4 has limited capacity to recruit neutrophils (Fig. 4e). Because CXCL7 activity depends on its N-terminal proteolytic processing, we also examined whether the surge in CXCL7 protein levels observed in injured muscles consists of proteolytically-cleaved (and hence active) CXCL7 and found it to be the case (Fig. 4f, g). On this basis, it is plausible that these platelet-specific chemokines are responsible for the recruitment of neutrophils to injured muscles, and that their reduced levels upon platelet depletion is the cause of defects in neutrophil infiltration.

## Neutrophil infiltration in regenerating muscles is reduced in Cxcl7KO mice

To test the hypothesis that platelet-specific chemokines regulate muscle regeneration by impacting neutrophil recruitment, we have utilized Cxcl7/Ppbp knockout (Cxcl7ko) mice[99], which have been found to exhibit normal platelet numbers and hemostatic functions (i.e., the platelet thrombotic response occurs normally upon activation)[99]. These mice have been obtained via the targeted deletion of the Cxcl7/Ppbp coding sequence[99] and, as expected, this leads to consequent loss of the plasma protein levels of CXCL7 (Fig. 5a). Although not directly targeted by the deletion of the Cxcl7/Ppbp coding sequence, the plasma levels of CXCL4/PF4 and CXCL5 also decline (Fig. 5a), presumably because these chemokines are encoded by genes that are located in adjacent loci and hence their expression is affected by deletion of the Cxcl7/Ppbp coding sequence[99]. On the other hand, the levels of CXCL1, which is encoded by a gene located in another genomic region, are not affected (Fig. 5a). Therefore, the Cxcl7ko mice that lack CXCL4, CXCL5, and CXCL7 provide a useful system to test the requirement of platelet-specific chemokines in neutrophil recruitment to injured muscles.

For these studies, we have used similar approaches as for the analysis of muscles from platelet-depleted mice (Fig. 2). Specifically, hematoxylin/eosin (H&E) staining of TA muscle sections from Cxcl7ko mice was utilized to determine the impact on muscle regeneration of loss of the platelet-specific chemokines CXCL4/5/7. Infiltration of immune cells in injured muscles was found, as expected, at day 1 after CTX injection in control mice but this was greatly reduced in the TA muscles from Cxcl7ko mice (Fig. 5b, c). Analyses of samples from 7 and 14 days after CTX injection further revealed that muscle regeneration is impeded by CXCL4/5/7 loss, as indicated by irregularities in the tissue ultrastructure (Fig. 5b, c).

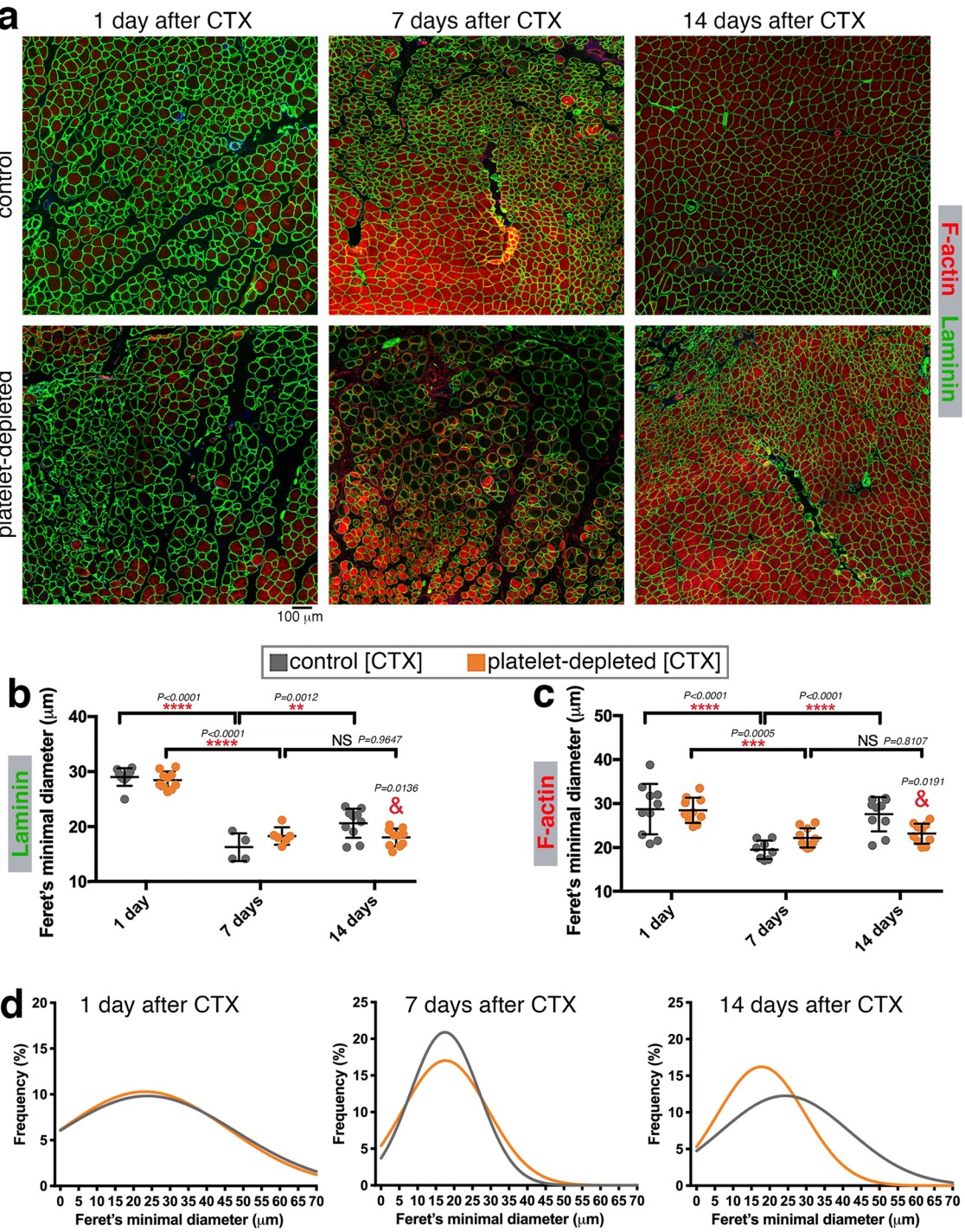

**Fig. 3 | Platelet depletion impedes the growth of myofibers during regeneration. a** Immunostaining of TA muscles from control and platelet-depleted mice at day 1, 7, and 14 from cardiotoxin (CTX)-induced injury with phalloidin (to detect F-actin; red) and with anti-Laminin antibodies (green) to detect the myofiber boundaries. **b, c** Quantitation of myofiber sizes (as estimated with the Feret's minimal diameter) based on anti-Laminin and phalloidin staining. Myofibers detected at day 1 largely consist of necrotic myofibers whereas myofibers found at day 7–14 are new myofibers resulting from de novo myogenesis. There are no significant changes in the size of myofibers found at day 7 in the muscles from platelet-depleted versus control mice, suggesting that platelet depletion does not impair myogenesis (see also Supplementary Fig. 4). However, myofiber size is significantly reduced at day 14 from CTX-induced injury in the muscles from platelet-depleted versus control mice. The graphs display the mean ±SD. In **b**, $n = 4$ (from 4 control independent mice) and $n = 6$ (from 6 platelet-depleted

independent mice) biologically independent CTX-injured muscles at day 7 after injury; and $n = 10$ biologically independent CTX-injured muscles obtained from $n = 10$ independent mice for each of the other timepoints and conditions analyzed. In **c**, $n = 8$ (from 8 control independent mice) biologically independent CTX-injured muscles at day 7 after injury; and $n = 10$ biologically independent CTX-injured muscles obtained from $n = 10$ independent mice for each of the other timepoints and conditions analyzed. **P < 0.01, ***P < 0.001, ****P < 0.0001 (two-way ANOVA with Tukey post hoc test); &P < 0.05 (two-way ANOVA with Sidak post hoc test) refers to the comparison of muscles from control versus platelet-depleted mice at day 14 of regeneration. **d** Gaussian plots that show the size range of all myofibers sourced from all TA muscles here analyzed, stained for F-actin. There are minimal changes in myofiber size (Feret's minimal diameter) at day 7 whereas platelet depletion leads to a significant reduction in myofiber size at day 14 from CTX injection. Source data are provided in the Source data file.

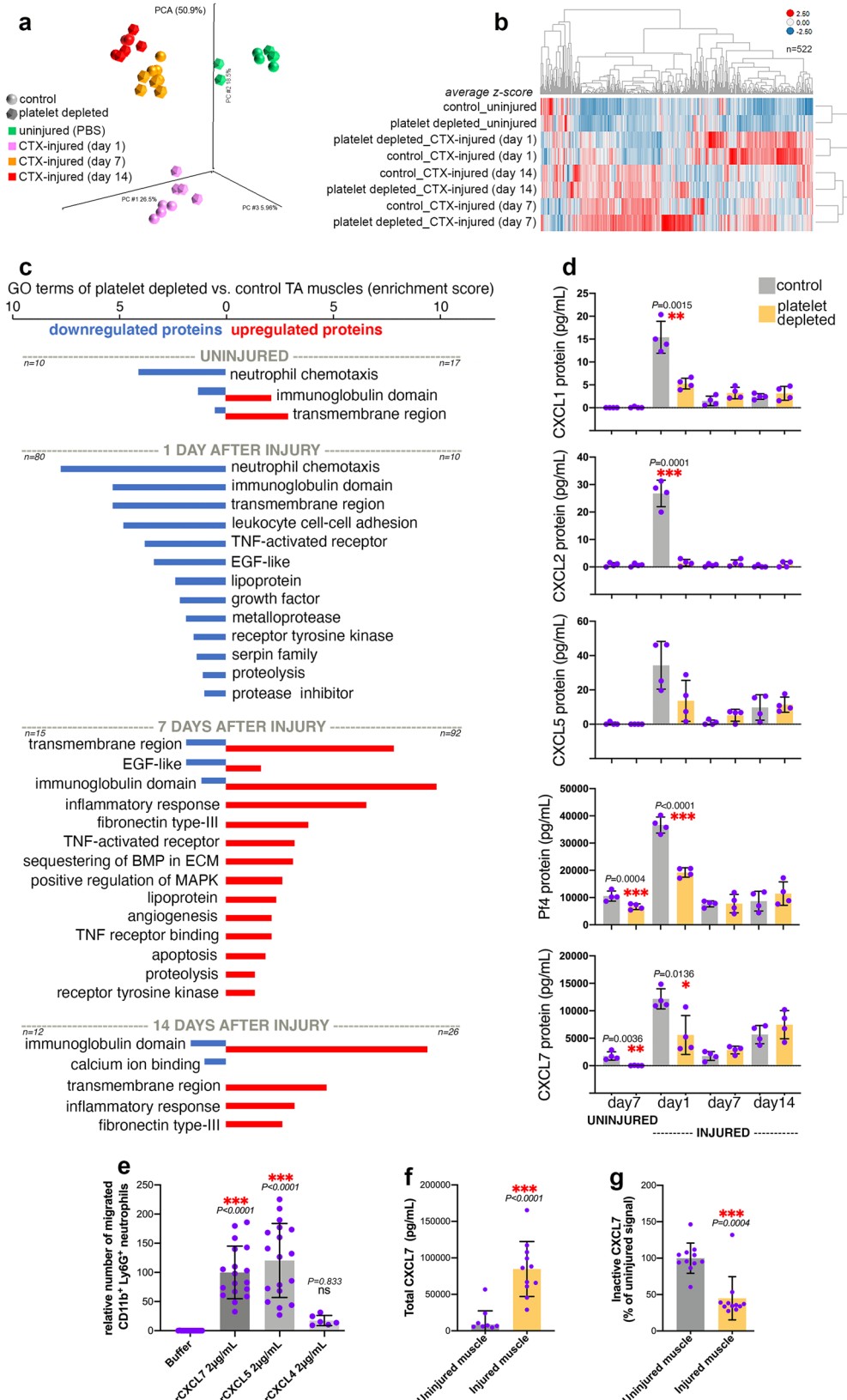

Immune cells invading the muscle at day 1 after CTX-induced injury primarily consist of neutrophils[2,3,11–13]. To test whether decreased recruitment of immune cells due to CXCL4/5/7 loss primarily results from reduced neutrophil recruitment, we have monitored via immunostaining the levels of the neutrophil markers MMP-9 and Ly6G (Fig. 5d, e). Co-staining with platelet markers revealed that neutrophils

can be found in close association with platelet thrombi (Fig. 5f). Loss of CXCL4/5/7 significantly reduced neutrophil recruitment to muscles at day 1 after CTX injection, compared to controls (Fig. 5d, e) and this was not due to changes in the amount of platelet thrombi detected in injured muscle (Fig. 5f). Altogether, these findings indicate that Cxcl7 knockout impairs neutrophil recruitment to injured skeletal muscles.

**Fig. 4 | Platelet depletion reduces the intramuscular levels of neutrophil chemoattractants in the early phase of muscle regeneration. a** Principal Component Analysis (PCA) of 640 cytokines profiled with Quantibody (Quantitative Multiplex ELISA) arrays from TA muscle homogenates obtained from mice with or without platelet depletion at 1, 7, and 14 days from CTX-induced injury. Uninjured muscles after 7 days from platelet depletion were also analyzed. **b** Heatmap of 522 cytokines that are prominently regulated (based on the average z-scores) during muscle regeneration and/or in response to platelet depletion. **c** Cytokine categories that are collectively regulated at different timepoints of muscle regeneration in a platelet-dependent manner include cytokines that promote neutrophil chemotaxis (day 1 from injury) and the inflammatory response (day 7–14 from injury). See also Supplementary Fig. 5. **d** Chemokines that promote neutrophil chemotaxis peak at day 1 after injury but their levels are reduced by platelet depletion. The graphs display the mean ±SD with $n = 4$ biologically independent muscles from 4 independent mice for each timepoint and condition; *$P < 0.05$, **$P < 0.01$, ***$P < 0.001$ (unpaired two-tailed $t$ test) refer to the comparison of muscles from control versus

platelet-depleted mice at a given timepoint. **e** In vitro neutrophil chemotaxis assays with recombinant versions of platelet-secreted chemokines. CXCL5 and CXCL7 have the strongest chemoattractant activity. The graphs display the mean ±SD with $n = 17$ (buffer), $n = 18$ (rCXCL7), $n = 18$ (rCXCL5), $n = 6$ (rCXCL4) biologically independent samples; ***$P < 0.001$ (one-way ANOVA with Tukey post hoc test), ns = not significant. **f** Consistent with the cytokine array data in **d**, ELISA assays indicate that the total levels of CXCL7 increase in skeletal muscle upon injury. The graph displays the mean ±SD with $n = 11$ biologically independent samples; ***$P < 0.0001$ (unpaired two-tailed $t$ test). **g** Additional ELISA assays with antibodies specific for inactive CXCL7 (i.e., which has not been proteolytically processed) indicate a decrease in inactive CXCL7 in injured muscles. The graph displays the mean ±SD with $n = 11$ biologically independent samples; ***$P = 0.0004$ (unpaired two-tailed $t$ test). Together, these data indicate that the surge in total CXCL7 observed in injured muscles largely consists of proteolytically-cleaved (and hence active) CXCL7. Source data are provided in the Source data file.

## Automated image analysis reveals a similar pattern of defective muscle regeneration in response to platelet depletion and Cxcl7ko

We next used image analysis based on machine learning to determine whether injured (CTX-injected) TA muscles from platelet-depleted and Cxcl7ko mice display similar defects in regeneration compared to control muscles. Automated analysis of H&E images reliably identified muscle-infiltrating immune cells, myofibers, and centrally-nucleated myofibers (Supplementary Fig. 7a), which are indicative of newly formed myofibers resulting from de novo myogenesis[2,3]. As expected, there were substantially no centrally-nucleated myofibers and minimal immune infiltration in uninjured (PBS-injected) TA muscles.

Compared to wild-type injured controls, TA muscles from platelet-depleted and Cxcl7ko mice displayed similar defects in immune cell infiltration at day 1 after injury (Supplementary Fig. 7b, d). These findings are consistent with the identification of defective neutrophil infiltration by immunostaining at this early stage of regeneration in the muscles from platelet-depleted and Cxcl7ko mice (Figs. 2–5).

The number of centrally nucleated myofibers is maximal at day 7 and then declines by day 14, a late stage at which the muscle has largely regenerated its mass and strength[3]. There was a trend towards decreased levels of centrally nucleated myofibers in the muscles from platelet-depleted and Cxcl7ko mice but these values were not statistically significant (Supplementary Fig. 7c, e), suggesting that myogenesis is not substantially affected by platelet-derived CXCL4/5/7. Altogether, these automated image analyses indicate that defective immune infiltration at day 1 after injury is a major and shared feature of altered regeneration in the muscles of platelet-depleted and Cxcl7ko mice.

## Defective neutrophil recruitment to injured muscles in Cxcl7ko mice leads to reduced myofiber size and muscle force production post-injury

We have found that platelets and the platelet-derived chemokines CXCL4, CXCL5, and CXCL7 guide neutrophil infiltration in the early steps of muscle regeneration (Figs. 1–5). In addition to cardiotoxin (CTX), intramuscular injection of glycerol is routinely used to induce skeletal muscle damage in mice[80,81,100–102]. Interestingly, glycerol elicits a stronger inflammatory response than CTX[102], suggesting that glycerol-induced injury may constitute a useful setting to study the impact of platelet-induced signaling on immune cell recruitment to injured muscles. On this basis, we next tested whether defective platelet-derived chemokine signaling impairs muscle regeneration also after glycerol-induced injury. To this purpose, we examined the TA muscles from WT and Cxcl7ko mice 14 days after glycerol-induced injury, a post-injury stage in which muscle regeneration is largely

completed[2,3,32]. In agreement with this model, all mice displayed recovery of the TA muscle mass and even post-injury muscle hypertrophy (Fig. 6a).

To examine the functional consequences of defective TA muscle regeneration in Cxcl7ko mice, we next measured the twitch force (i.e., physiological, spontaneous-like force) generated by WT and Cxcl7ko mice 14 days after injury. Consistent with the overall completion of regeneration at this stage[2,3,32], there was no significant difference in the twitch force of post-injury versus uninjured WT muscles (Fig. 6b). However, TA muscles from Cxcl7ko mice displayed significant reduction in the twitch force compared to uninjured controls, indicating that muscle regeneration is incomplete, presumably due to impairment of signaling by platelet-specific CXCL4/5/7 chemokines (Fig. 6b). Analysis of muscle fatigability, which is based on a series of maximal (tetanic) stimulations of TA muscles, revealed no difference post-fatigue but confirmed that post-injury TA muscles from Cxcl7ko mice but not from WT mice display reduced force production in the pre-fatigue state (Fig. 6c).

To understand the mechanistic basis of decreased muscle force in Cxcl7ko mice post-injury, we analyzed the size of distinct myofiber types in TA muscles from WT and Cxcl7ko mice, at 14 days post-injury or uninjured (Fig. 6d). These histological analyses highlight that regeneration is defective in post-injury muscles from Cxcl7ko mice, as indicated by the presence of debris and empty spaces in-between the myofibers, compared to post-injury muscles from isogenic control mice (Fig. 6d). Moreover, the Gaussian plots (representing all myofibers sourced from all TA muscles in each group) indicate that the Feret's minimal diameter of type 2a and type 2x myofibers is reduced in Cxcl7ko mice at 14 days post-injury (Fig. 6e, f) whereas the size of type 2b myofibers is not affected compared to post-injury WT muscles (Fig. 6g).

The analysis of the myofiber size across individual TA muscles led to overall similar conclusions: there was no significant difference in the size of type 2a, 2x, and 2b myofibers when comparing uninjured versus post-injury WT muscles (Fig. 6h). However, there was a significant decline in the size of myofibers in post-injury versus uninjured muscles from Cxcl7ko mice (Fig. 6h). Such decline in myofiber size was not accompanied by any major changes in the relative proportion of different myofiber types that compose the TA muscle (Fig. 6i). Moreover, there was a higher myofiber number in both WT and Cxcl7ko post-injury muscles (Fig. 6j), consistent with previous studies that have examined TA muscle regeneration[2,3,32], although such increase was significant only for WT muscles (Fig. 6j). We also analyzed the intramuscular fat infiltration, which occurs in post-injury muscles in particular after glycerol-induced injury[80,81,100,102], but found no difference when comparing muscles from WT versus Cxcl7ko mice (Supplementary Fig. 8).

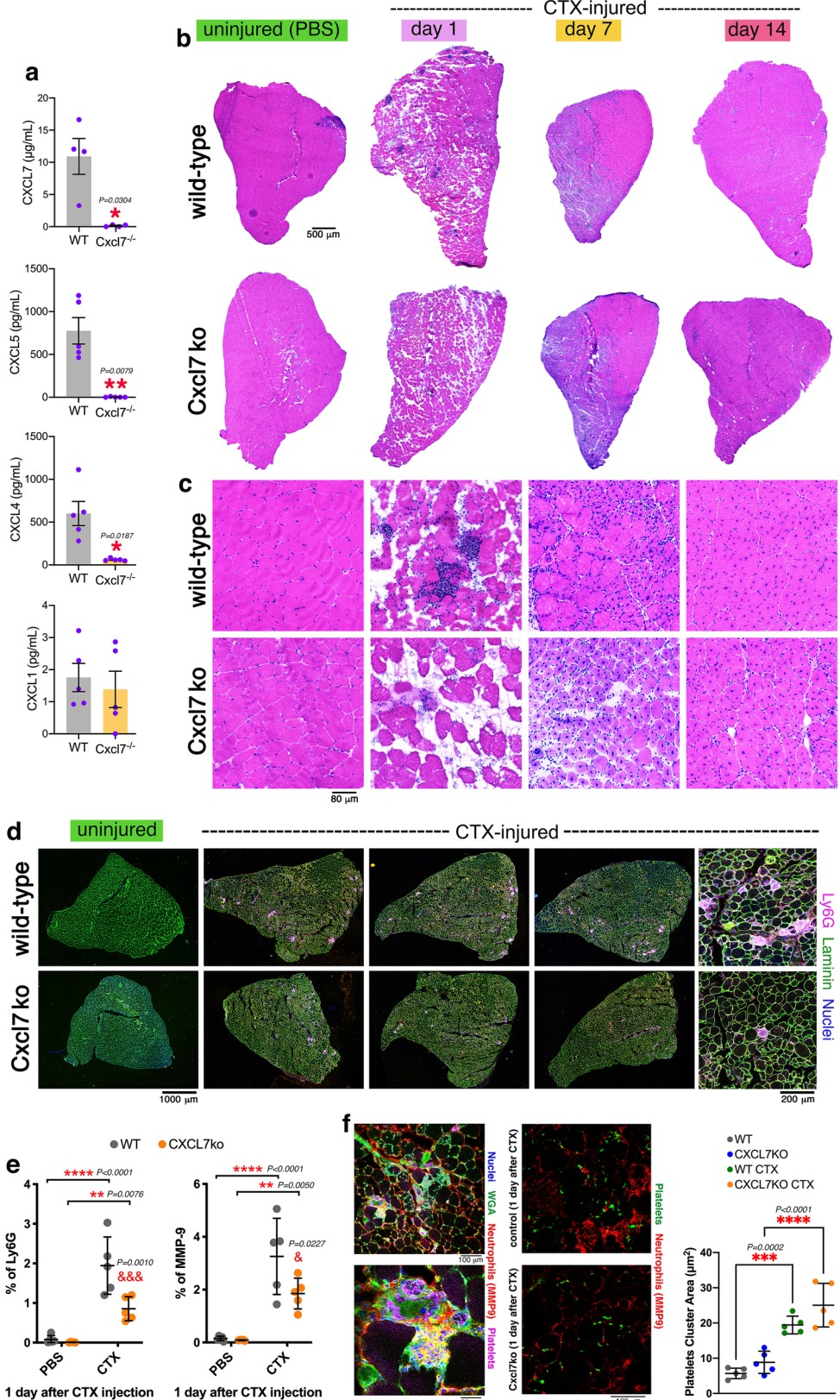

Altogether, these studies indicate that physiological force production by skeletal muscles from Cxcl7ko mice is impaired post-injury due to reduced myofiber size, and that platelet-derived CXCL4/5/7 chemokine signaling is an early step of muscle regeneration that is key for the re-establishment of skeletal muscle function.

## Neutrophil depletion impairs myofiber growth and neo-angiogenesis during muscle regeneration

Previous studies have found that neutrophils are necessary for muscle regeneration[2,3,18]. Here, we have found that platelet depletion and Cxcl7 knockout impair neutrophil recruitment to injured muscles and that this in turn impedes myofiber growth and force production in

**Fig. 5 | Neutrophil infiltration in injured muscles is impeded by Cxcl7ko platelets. a** Plasma levels of the neutrophil chemoattractant CXCL7 are reduced in Cxcl7ko (Cxcl7$^{-/-}$) mice. CXCL5 and CXCL4 (encoded by adjacent genes) are also reduced whereas CXCL1 is not affected. The graphs display the mean ± SD with $n = 4$ (CXCL7) and $n = 5$ (CXCL1, CXCL4, CXCL5) biologically independent samples from $n = 4$ and $n = 5$ independent mice, respectively; *$P < 0.05$ and **$P < 0.01$ (unpaired two-tailed $t$ test). **b, c** H&E staining of TA muscles from control and Cxcl7ko mice at day 1, 7, and 14 from cardiotoxin (CTX) injection, and uninjured. Immune infiltration at day 1 after CTX-mediated injury is reduced in the muscles from Cxcl7ko mice. Ultrastructural defects can be seen at day 14 post-injury in muscles from Cxcl7ko versus control mice. **d** Immunostaining of muscles from control and Cxcl7ko mice for neutrophil markers, i.e., Ly6G (purple). Myofiber boundaries are identified with immunostaining for anti-Laminin antibodies (green) whereas nuclei are stained with DAPI (blue). **e** Intramuscular neutrophil infiltration at day 1 from injury is significantly reduced in Cxcl7ko mice. Similar results are found with both Ly6G and MMP9 immunostaining. The graphs display the mean ±SD with $n = 5$ biologically independent samples for each group and condition from $n = 5$ independent mice; **$P < 0.01$, ***$P < 0.001$, ****$P < 0.0001$ (two-way ANOVA with Tukey post hoc test); $^{\&}P < 0.05$ and $^{\&\&\&}P < 0.001$ (two-way ANOVA with Sidak post hoc test) refer to the comparison of CTX-injured WT and Cxcl7ko at a given timepoint of regeneration. **f** Normally, platelets (green) are found in association with neutrophils (red) in injured muscles. Although lack of CXCL7 decreases neutrophil recruitment, the area of platelet thrombi that is found in injured muscles of WT and Cxcl7ko mice is similar, indicating that defective neutrophil recruitment does not result from lower platelet number or aggregation in Cxcl7ko mice, consistent with previous studies that have found that platelet numbers and hemostatic functions are not impaired in these mice[99]. The graph displays the mean ±SD with $n = 5$ biologically independent samples for each group and condition from $n = 5$ independent mice; ***$P < 0.001$ (two-way ANOVA with Tukey post hoc test). Source data are provided in the Source Data file.

regenerating muscles (Figs. 2–6). To test whether neutrophils are indeed necessary for muscle regeneration, we have utilized anti-Gr1 and anti-Ly6G antibodies to deplete neutrophils and compared these mice to IgG-treated controls. For each of these mice, one leg was injured with glycerol whereas the contralateral leg was injected with PBS (control).

Histological analyses of tibialis anterior muscles at 10 days post-injury indicate that myofiber size is reduced in the muscles of neutrophil-depleted versus mock-treated mice whereas the uninjured muscles are not affected by neutrophil depletion (Fig. 7a). Gaussian curves (obtained from the cumulative analysis of all myofibers in a group of muscles) indicate that the size of type 2a, 2x, and 2b myofibers is reduced in post-injury muscles upon neutrophil depletion compared to IgG control treatments whereas there is no effect of neutrophil depletion on the size of myofibers in the contralateral uninjured muscles (Fig. 7b–d). Analysis of the average myofiber size for each mouse indicates that neutrophil depletion significantly reduces the size of type 2x ($P = 0.019$) and type 2b ($P = 0.0036$) myofibers in post-injury muscles compared to mock-treated controls and that the size of type 2a myofibers is also reduced, albeit with $P = 0.0570$ (Fig. 7e). There are no major effects on the myofiber type distribution, apart for an increase in the percentage of type 2x myofibers which occurs in both neutrophil-depleted and controls (Fig. 7f). Lastly, these histological analyses indicate that there is an overall increase in the number of myofibers in post-injury muscles from both neutrophil-depleted and control mice (Fig. 7g). Altogether, these findings indicate that neutrophil depletion impairs myofiber growth during muscle regeneration.

In addition to removing debris and promoting the organized recruitment of other immune cell types, neutrophils promote tissue regeneration also via the secretion of many signaling factors, which include regulators of angiogenesis such as the vascular endothelial growth factor (VEGF)[30,82,103–107]. Moreover, neutrophils express high levels of MMP9 and other metalloproteases that release additional VEGF bound to the extracellular matrix, further enhancing VEGF bioavailability[28,82]. On this basis, we have examined the abundance of capillaries (based on PECAM-1 staining) and found that neutrophil depletion significantly reduces neo-angiogenesis in post-injury muscles compared to mock-treated controls (Supplementary Fig. 9), as also observed upon platelet depletion (Supplementary Fig. 6). Moreover, analysis of fat infiltration (identified by perilipin-1 staining) indicates a trend towards an increase (Supplementary Fig. 9). Coincident with the decline in myofiber size and neo-angiogenesis, post-injury muscles from mice with neutrophil depletion produce a significantly lower normalized twitch and tetanic force compared to mock-treated controls whereas neutrophil depletion does not impact force production in the contralateral uninjured muscles (Fig. 7h, i).

Altogether, these studies indicate that the recruitment of neutrophils to injured muscles is necessary for optimal blood vessel formation, myofiber growth, and force production. On this basis, defective neutrophil recruitment appears to be a primary reason for the impairment of muscle regeneration in response to platelet depletion and knockout of platelet-derived Cxcl7.

## Discussion
Skeletal muscle has the remarkable capacity to regenerate in response to injuries caused by many physiological and pathological insults[1–5]. The sequential recruitment of immune cell populations and the de novo formation of myofibers are key events in muscle regeneration that have been extensively studied[2,3,11–13]. However, less is known about the early steps of muscle repair and the consequences of their derangement on later phases of muscle regeneration and post-injury muscle function.

Platelets are among the first responders to tissue injuries[108] but their role in muscle regeneration has not been explored. In addition to their classical roles in hemostasis, there is growing appreciation that platelets have important signaling functions[109–111]. For example, our previous work has found that platelets promote cancer cell metastasis via secretion of TGF-β and other pro-metastatic signaling factors that induce epithelial-to-mesenchymal transition and migration of cancer cells[73,112–114]. In this study, we have found that platelets guide the early steps of muscle regeneration by promoting the recruitment of neutrophils to injured muscles: platelet-specific chemokines (CXCL5 and CXCL7[94–98]) are necessary for the infiltration of neutrophils into injured muscles, and impairment of such platelet-derived chemokine signaling impedes this process.

Platelet localization and neutrophil recruitment to muscles is maximal within the first 24 h after injury and then declines in the subsequent days (Figs. 1 and 2). Likewise, the levels of the platelet-secreted neutrophil chemoattractants CXCL5 and CXCL7 are maximal at day 1 after injury (Fig. 4), indicating that platelet-initiated intercellular signaling occurs in the early steps of muscle regeneration, presumably immediately upon platelet activation and the formation of thrombi in injured muscles.

Interestingly, derangement of this early step of muscle regeneration has consequences at later stages. Specifically, platelet depletion and Cxcl7 knockout leads to the regeneration of muscles that have decreased myofiber size (Figs. 3 and 6), as also found with neutrophil depletion (Fig. 7). As estimated with the analysis of eMHC levels (Supplementary Fig. 4), myogenesis seems not be affected by platelet depletion, suggesting that the growth rather than the de novo formation of myofibers is impacted by platelets. However, this occurs in the late phases of muscle regeneration, a stage at which neutrophils have migrated out of the muscles and the levels of platelet-specific secreted factors (defined based on the platelet proteome[115]) detected in muscles are minimal (Fig. 4). There are however several cytokines and growth factors that are differentially modulated in the late phase of muscle regeneration when comparing platelet-depleted versus control mice. These include cytokines such as tumor necrosis factor ligands (Fig. 4 and Supplementary Fig. 5) that have been found to promote myogenesis but that are known inducers of myofiber atrophy

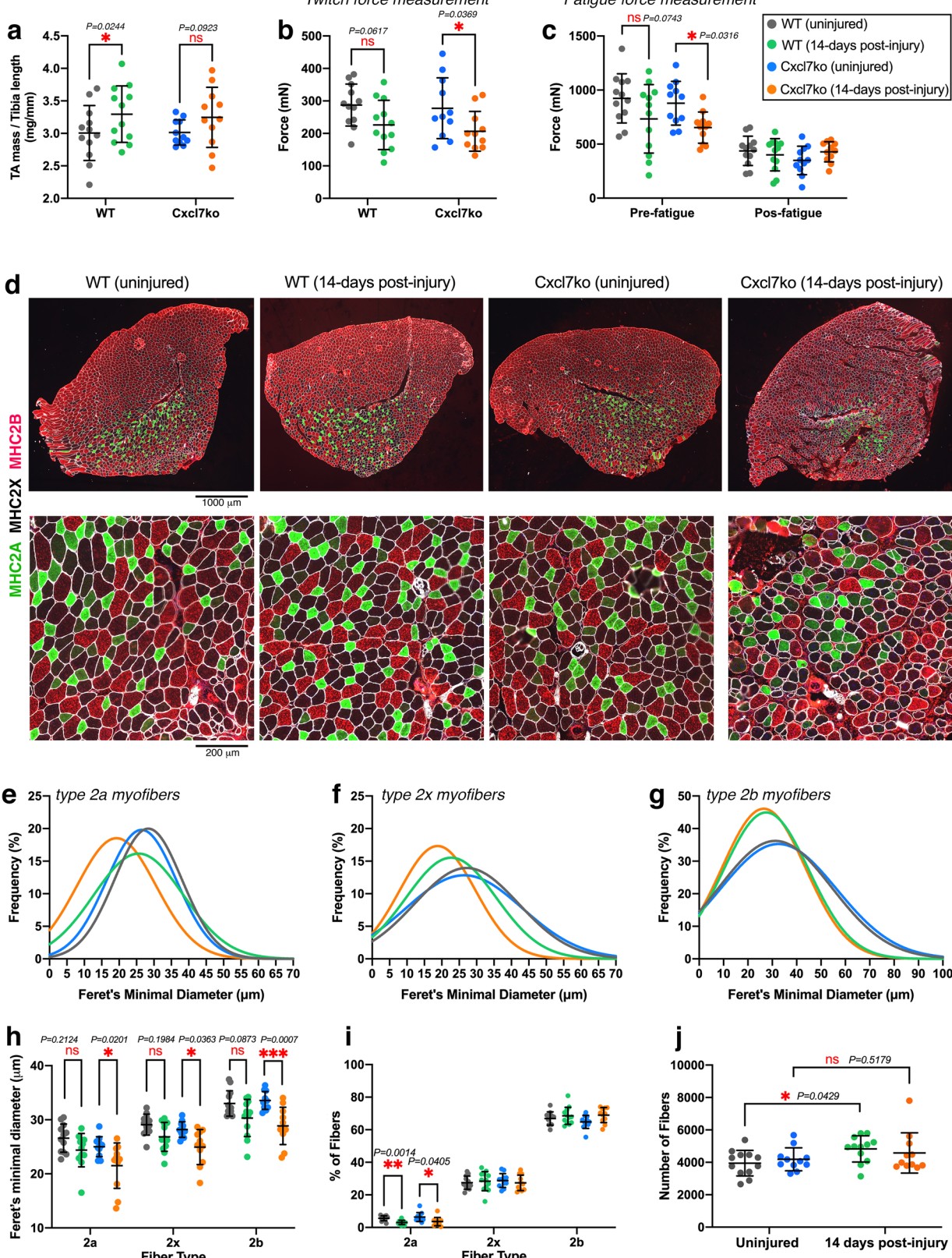

during cancer cachexia and in other disease states that promote muscle wasting[88–91]. These factors are likely contributed by muscle-infiltrating cells other than platelets and neutrophils, such as macrophages, for which we find increased recruitment to injured muscles of platelet-depleted mice at day 7 from injury, compared to controls (Fig. 2f and Supplementary Figs. 2 and 3). Therefore, a plausible model

is that defective platelet-derived chemokine signaling and reduced neutrophil recruitment to injured muscles in the early steps of muscle regeneration leads to unresolved tissue damage and consequent excessive inflammation and macrophage recruitment at later stages of regeneration, and that this stunts the growth of new myofibers because of the high levels of atrophic ligands. Moreover, we find that

**Fig. 6 | Myofiber size and force production are reduced in post-injury muscles from Cxcl7ko mice.** Analysis of TA muscles after 14 days from glycerol-induced injury. **a** TA muscle weight normalized by the tibia bone length indicates that there is post-injury hypertrophy, although this is not significant for Cxcl7ko mice. **b** There is no significant difference in the twitch force of uninjured and post-injury muscles from WT mice, indicating that regeneration has recovered muscle function. However, post-injury muscles from Cxcl7ko mice are significantly weaker compared to uninjured muscles. **c** Similar deficits in muscle force are found in pre-fatigue muscles from Cxcl7ko mice, whereas there are no differences post-fatigue. In **a**–**c**, the graphs display the mean ±SD with $n = 12$ (WT) and $n = 11$ (Cxcl7ko) biologically independent samples from $n = 12$ and $n = 11$ independent mice, respectively; *$P < 0.05$ (two-way ANOVA with Tukey post hoc test), ns = not significant. **d** Immunostaining of uninjured and post-injury muscles from control and Cxcl7ko mice with antibodies for myosin heavy chain isoforms to detect type 2a myofibers (green), type 2x myofibers (black), and type 2b myofibers (red). Defects in

regeneration (such as space in-between myofibers) are found in post-injury muscles from Cxcl7ko mice compared to post-injury controls. **e**–**g** Gaussian plots indicate an overall reduced size (Feret's minimal diameter) of type 2a (**e**) and type 2x (**f**) myofibers in post-injury muscles from Cxcl7ko mice, compared to post-injury muscles from WT mice and uninjured controls. **h** Quantitation of myofiber sizes based on the average values obtained from the individual muscles in a group. There is a significant decline in the size of myofibers in post-injury muscles from Cxcl7ko mice whereas myofiber size differences between uninjured and post-injury WT muscles are not significant. **i** There is an overall similar myofiber type composition of uninjured and post-injury TA muscles. **j** Myofiber number similarly increases in post-injury versus uninjured muscles from WT and Cxcl7ko. In **h**–**j**, the graphs display the mean ±SD with $n = 12$ (WT) and $n = 11$ (Cxcl7ko) biologically independent samples from $n = 12$ and $n = 11$ independent mice, respectively; *$P < 0.05$, **$P < 0.01$, ***$P < 0.001$, ns = not significant (two-way ANOVA with Tukey post hoc test). Source data are provided in the Source data file.

platelet depletion decreases VEGF levels in the early phase of muscle regeneration (Supplementary Fig. 6) and that neo-angiogenesis is impeded in post-injury muscles from mice with platelet depletion (Supplementary Fig. 6) and in mice with neutrophil depletion (Supplementary Fig. 9). Because neo-angiogenesis precedes myogenesis during regeneration[116–118], defective neo-angiogenesis likely contributes to the impediment of myofiber growth observed upon depletion of neutrophils, platelets, and platelet-secreted chemokines.

In addition to promoting the removal of cellular debris and to setting the stage for the infiltration of other immune cells[2,3,11–13,18], neutrophils also influence the metabolic capacity of skeletal muscle[119]. Specifically, it was previously found that neutrophils support muscle force production via secretion of IL-1β, which promotes muscle performance by priming exercise-dependent GLUT4 translocation and glucose metabolism[119]. We have found that impaired neutrophil recruitment due to CXCL7 loss results in decreased muscle force production post-injury. While this decreased muscle performance likely stems from reduced myofiber growth, it may also arise from derangement of neutrophil signaling to muscle satellite cells, and the consequent negative impact on muscle metabolism.

In addition to physiological repair, muscle regeneration is also altered in several diseases, such as muscular dystrophy, which is characterized by damage-regeneration cycles that ultimately lead to stem cell depletion and to the incapacity to repair skeletal muscles[7]. Specifically, the lack of dystrophin leads to membrane tears and a rise in $Ca^{2+}$ levels which are ultimately responsible for myofiber necrosis in Duchenne muscular dystrophy[7]. Interestingly, previous studies have found that platelet adhesion and aggregation are defective in patients with Duchenne muscular dystrophy[120–122]. On this basis and considering our finding that platelets contribute to skeletal muscle regeneration, it is possible that impairment of platelet function contributes to the chronic inflammation and to the defective regeneration of dystrophic muscles, as we have observed in this study upon experimental depletion of platelets (Figs. 2 and 3). Future studies should address whether injection of functional platelets and/or recombinant CXCL7 can aid regeneration and reduce chronic inflammation in skeletal muscles of mouse models and patients with Duchenne muscular dystrophy. Likewise, injection of platelets or and/or recombinant CXCL7 may aid muscle regeneration by boosting neutrophil recruitment in the context of aging and age-related diseases such as diabetes, which are characterized by decreased regenerative capacity[10,123–125].

Platelet-released chemokines may also help the regeneration of more severe injuries, such as volumetric muscle loss (VML), which consists in the quick loss of >20% muscle mass[126,127]. In this context, the signaling interactions between platelets and neutrophils and/or recombinant platelet-secreted chemokines (e.g., CXCL5/7) may help the regeneration of VML injuries by promoting the efficacy of currently-used interventions such as tissue and stem cell engraftment[126–128]. Conversely, limiting platelet-derived chemokine

signaling could help prevent excessive neutrophil infiltration, which occurs in unresolved VML injuries and contributes to the impairment of muscle stem cell function[129,130].

There is also growing interest in the possibility of using platelets for drug delivery[131–133]. For example, because of their capacity to interact with cancer cells, doxorubicin-loaded platelets have been used to selectively target cancer cells while reducing general doxorubicin toxicity[134]. On this basis and considering our finding that platelets are recruited to regenerating skeletal muscles, we propose that platelets might be employed as drug carriers to deliver pro-regeneration factors specifically to injured skeletal muscles.

Altogether, this study indicates that platelet-initiated chemokine signaling guides the early steps of muscle regeneration by promoting neutrophil recruitment and that this in turn impacts myofiber size and muscle strength post-injury (Fig. 8). We propose that platelet-derived chemokines may provide therapeutic opportunities for promoting muscle regeneration.

## Methods

### Mouse husbandry
All mice were housed in the Animal Resource Center at St. Jude Children's Research Hospital, fed a standard chow diet, and handled in accordance with protocols approved by the St. Jude Children's Research Hospital Institutional Animal Care and Use Committee (IACUC). Additional accreditation of the Animal Resource Center at St. Jude Children's Research Hospital was provided by the Association for Assessment and Accreditation of Laboratory Animal Care (AAALAC). Mice were housed in a ventilated rodent-housing system with a controlled temperature (22–23 °C), 40% humidity, 12-h light/dark cycle, and given free access to food and water. Humane endpoints were not exceeded in any experiment. Euthanasia was performed with carbon dioxide in agreement with the recommendations of the Panel on Euthanasia of the American Veterinary Medical Association.

### Mouse models
C57BL6/J male mice (The Jackson Laboratory, JAX#000664) were utilized at 5–6 months of age, a timepoint at which postnatal skeletal muscle growth has halted. CXCL7KO mice were previously described[99]: also in this case, male mice were utilized at 5–6 months of age. Littermate wild-type males were used as controls in experiments with CXCL7KO mice, which were genotyped before experimental use by Transnetyx. BALB/c male mice (The Jackson Laboratory, JAX#000651) were utilized at 3 months of age for experiments with neutrophil-depleting antibodies and the corresponding IgG2a/b control antibodies.

### Antibody-mediated platelet depletion
For platelet depletion experiments, mice were injected via the tail vein with a single dose (100 μg per mouse) of a platelet-depleting antibody

 

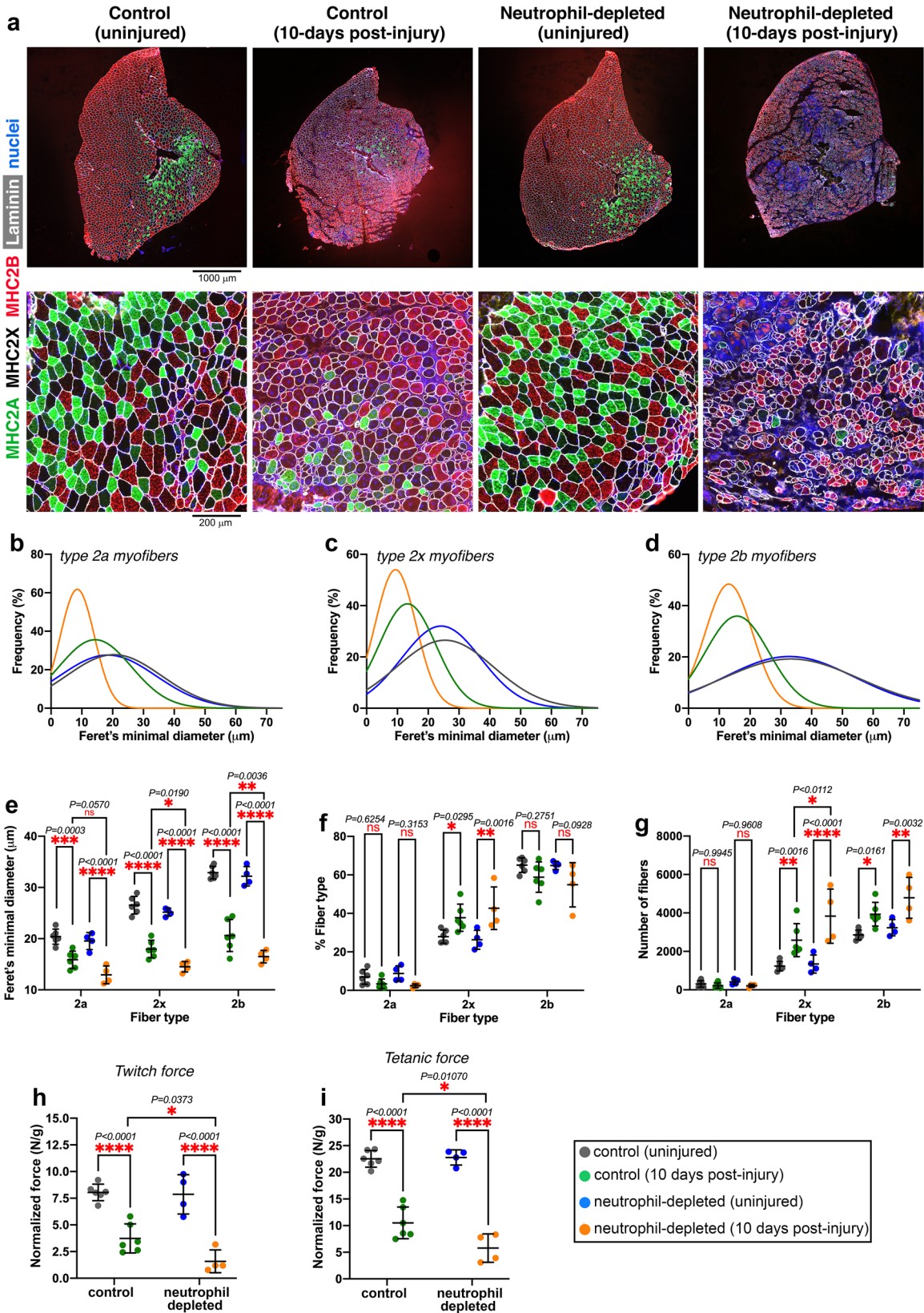

(anti-GP1bα; R300, Emfret) or IgG control (R301, Emfret) 2 h before CTX injection into the TA muscle. Platelet-depletion with this anti-GP1bα antibody leads to a greater than 95% reduction in platelet counts, which is achieved within 60 min of antibody injection[71–74]. At day 4 from CTX injection, the mice were injected via the tail vein with a second dose of platelet-depleting antibody or IgG control.

**Antibody-mediated neutrophil depletion**

For neutrophil depletion, mice were injected via the tail vein with 200 μg per mouse of a neutrophil-depleting antibody (anti-Gr1; Bio-Legend, clone RB6-8C5) or the isotype control (IgG2b; BioXCell, clone LTF-2) 24 h before glycerol-mediated injury of the TA muscle. Subsequently, these mice were further injected by i.p with another

**Fig. 7 | Neutrophil depletion impairs skeletal muscle regeneration.** Analysis of TA muscles after 10 days from glycerol-induced injury. **a** Immunostaining for myosin heavy chain isoforms was utilized to detect type 2a (green), type 2x (black), and type 2b myofibers (red): these histological analyses indicate defective muscle regeneration in mice with neutrophil depletion. **b–d** Gaussian plots indicate that neutrophil depletion impairs the growth of newly formed myofibers in post-injury muscles whereas there is no effect of neutrophil depletion on myofiber size in contralateral uninjured muscles. **e** Quantitation of myofiber sizes (Feret's minimal diameter) based on the average values obtained from the individual muscles in a group. There is an overall significant decline in the size of type 2x and 2b myofibers in post-injury muscles from neutrophil-depleted mice compared to post-injury muscles from mock-treated mice. **f** There is an overall similar myofiber type composition of uninjured and post-injury TA muscles. However, post-injury muscles (both from neutrophil-depleted and mock-treated mice) display higher levels of type 2x myofibers. **g** The number of type 2x and 2b myofibers increases in post-

injury versus uninjured muscles and the number of type 2b myofibers is significantly higher in the muscles from neutrophil-depleted mice. In **e**–**g**, the graphs display the mean ±SD with $n = 6$ (from 6 independent control mice) and $n = 4$ (from 4 independent neutrophil-depleted mice) biologically independent muscles; *$P < 0.05$, **$P < 0.01$, ***$P < 0.001$, ****$P < 0.0001$ (two-way ANOVA with Tukey post hoc test), ns = not significant. **h** The normalized twitch force of post-injury muscles from neutrophil-depleted mice is reduced compared to that of post-injury muscles from mock-treated mice. **i** Similar deficits in muscle force production are found for the normalized tetanic force of muscles from neutrophil-depleted versus mock-treated mice. Neutrophil depletion does not impact twitch and tetanic force production by uninjured muscles (**h**, **i**). In **h**, **i**, the graphs display the mean ± SD with $n = 6$ (from 6 independent control mice) and $n = 4$ (from 4 independent neutrophil-depleted mice) biologically independent muscles; *$P < 0.05$, ****$P < 0.0001$ (two-way ANOVA with Sidak post hoc test). Source data are provided in the Source data file.

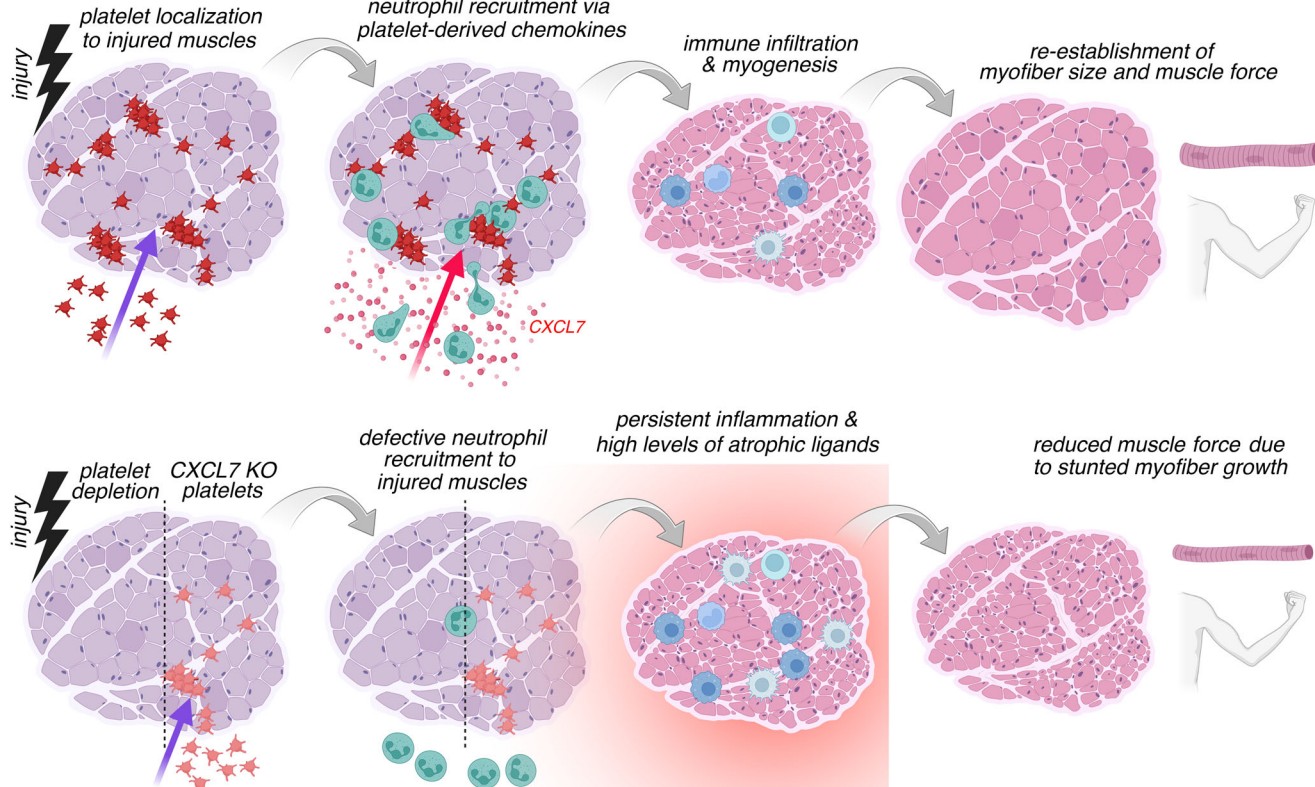

**Fig. 8 | Platelets promote skeletal muscle regeneration by guiding the recruitment of neutrophils to injured muscles via the platelet-released chemokine CXCL7.** In response to injury, platelets localize to and form thrombi in skeletal muscles and promote the recruitment of neutrophils via the release of platelet-specific chemokines (e.g., CXCL7) that are neutrophil chemoattractants. Neutrophil infiltration is known to promote muscle repair via the removal of cellular debris and by setting the stage for the subsequent steps of regeneration, which include the infiltration of monocytes and macrophages and myogenesis, i.e., the de novo formation of myofibers. In the absence of platelets or when platelets lack CXCL7 (CXCL7KO), the recruitment of neutrophils to injured muscles is

defective. This in turn leads to unresolved tissue damage, excessive recruitment of macrophages at later phases of regeneration, and to high levels of atrophic ligands that stunt the growth of newly-formed myofibers. Neo-angiogenesis is also reduced. Consequently, post-injury muscles arising from regeneration in the absence of early-stage platelet-initiated chemokine signaling display reduced myofiber size and lower muscle force production. Similar results are found with the experimental depletion of neutrophils. Altogether, these findings indicate a key role for platelet-induced chemokine signaling in ensuring optimal muscle regeneration by guiding the recruitment of neutrophils to injured muscles in the early phase after injury.

neutrophil-depleting antibody (anti-Ly6G; BioXCell, clone 1A8, 200 μg per mouse) or the corresponding isotype control (IgG2a; clone 2A3, 200 μg per mouse) 48 h and 96 h after the first dose of neutrophil-depleting antibody. BALB/c male mice (The Jackson Laboratory, JAX#000651) at 3 months of age were utilized for these experiments because anti-Gr1 and anti-Ly6G antibodies work better in this strain to deplete neutrophils[73,135–137].

## Skeletal muscle injury protocols

For skeletal muscle injury with cardiotoxin (CTX), 50 μL of CTX (cardiotoxin from *Naja mossambica mossambica*, Sigma cat. no. C9759) was injected into the tibialis anterior (TA) muscle at a concentration of 0.3 mg/mL in PBS whereas the TA in the contralateral leg was mock-injected with PBS. For skeletal muscle injury with glycerol, 70 μL of 50% glycerol (Sigma, cat. no. G5516) was injected into the tibialis anterior

muscle whereas the TA in the contralateral leg was mock-injected with PBS. Subsequently, the TA muscles were excised and frozen in iso-pentane cooled with liquid nitrogen for histology after 1, 7, and 14 days from injury, and snap-frozen in liquid nitrogen for the preparation of tissue homogenates for cytokine arrays.

## Immunostaining on tibialis anterior skeletal muscle sections

For immunostaining, TA muscles were bisected at the mid-belly, mounted onto tragacanth gum, and frozen in liquid nitrogen-cooled isopentane (Sigma-Aldrich, cat. no. 277258); 10-μm sections were cut on a cryostat and immunostained as previously done[138–141]. Unfixed slides holding the sections were incubated with blocking buffer (PBS with 0.1% Triton X-100, 1% BSA, and 2% horse serum) for 1 h before incubation with primary antibodies, which were all used at 1:150 for immunostaining.

For immunostaining muscle-infiltrating immune cells, the fol-lowing antibodies were used: anti-MMP9 (R&D Systems, cat. no. AF909) and anti-Ly6G (BioLegend, cat. no. 17-9668-82) to detect neutrophils; anti-F4/80 (BioLegend, cat. no. 123119) to immunostain total macrophages, anti-CD68 to immunostain M1 macrophages (Abcam, cat. no. ab125212), and anti-CD206 (Macrophage Mannose Receptor; 6068c2, Biolegend, cat. no. 141711) to immunostain M2 macrophages; and anti-GP1bβ antibodies to detect platelets (Emfret, cat. no. X649). In addition, rat anti-laminin α2 antibodies (4H8-2; Santa Cruz, cat. no. sc-59854) or WGA (Wheat Germ Agglutinin, Alexa Fluor 555 conjugate, ThermoFisher, cat. no. W32464) were used to delineate the myofiber boundaries. Anti-eMHC antibodies were used to detect embryonic myosin heavy chain (anti-MYH3, Santa Cruz, cat. no. SC-5309). Anti-PECAM-1 antibodies (MEC13.3; BD Biosciences) were uti-lized to identify blood vessels. Immunostaining for Perilipin-1 (Cell Signaling Technologies, cat. no. 9349) was used to identify fat infiltration in skeletal muscles after regeneration[81]. Nuclei were detected with DAPI (4′,6-diamidino-2-phenylindole; Roche, cat. no. 10236276001). For the analysis of immune cell infiltration, the images were threshold-adjusted and the percentage of the muscle field area occupied by immune cells was calculated by using the Nikon Elements software (version 4.11.0). All images within an experiment were pro-cessed equally.

For myofiber size and type analysis, TA muscle sections were incubated with antibodies against type 2 A (DSHB, cat. no. SC-71) and 2B myosin heavy chain (DSHB, cat. no. BF-F3) and rat anti-laminin α2 (4H8-2; Santa Cruz, cat. no. sc-59854) overnight at 4 °C. The sections were then washed and incubated with secondary antibodies for type 2A (anti-mouse IgG1 Alexa488, Life Technologies cat. no. A21121), type 2B (anti-mouse IgM Alexa555, Life Technologies cat. no. A21426), and laminin (anti-rat IgG Alexa647, Life Technologies cat. no. A21247). The whole tibialis anterior section was imaged on a Nikon C2 confocal microscope with a ×10 objective and stitched to compile an overview of the muscle. The myofiber types and sizes were analyzed with the Nikon Elements software (version 4.11.0) by using the inverse thresh-old of laminin α2 staining to determine myofiber boundaries. The myosin heavy chain staining was used to classify type 2B myofibers (red), type 2A (green), and presumed 2× myofibers (black) that were not stained for 2B or 2A. After the myofibers were classified and the parameters measured, the Feret's minimal diameter was used as measurement of the myofiber size due to its accuracy in estimating the size of unevenly shaped or cut objects.

For the quantification of the number of myofibers, all myofibers in the cross-sections of entire tibialis anterior muscles were counted based on the myofiber borders identified by laminin immunostaining. The size and number of myofibers were measured from the inverse images of laminin immunostaining (for identifying myofiber borders), excluding myofibers with diameters <2 and >100 μm. To categorize myofiber types, the intersections of the inverse images of laminin and myosin heavy chain-specific staining were used. These analyses were performed using the Nikon Elements software (version 4.11.0) and the "Object count" function.

## Hematoxylin and eosin (H&E) staining

Frozen TA muscle sections (10 μm-thick) were mounted on positively charged glass slides (Superfrost Plus; Thermo Fisher Scientific, Wal-tham, MA), and dried at room temperature for 1 h. Tissue sections were then stained with H&E according to standard procedures[142].

## Schemes

Schemes were drawn with BioRender.

## Muscle force measurements

The measurement of the twitch and tetanic force of the tibialis anterior (TA) muscle was done as previously described[139,140,143] and normalized by the TA mass. Mice were deeply anesthetized via isoflurane and monitored throughout the experiment. The distal tendon of the tibialis anterior was carefully dissected and individually tied with braided surgical silk (CynaMed Suture Thread with Needle; 12, 5/0, 19 mm Blade, 1/2 Reverse Cutting). The sciatic nerve was exposed and all branches were cut except for the common peroneal nerve. The foot was secured to a platform and the knee immobilized using a stainless-steel pin. The body temperature was monitored and maintained at 37 °C. The suture from the tendon was individually attached to the lever arm of a 305B dual-mode servomotor transducer (Aurora Sci-entific, Ontario, Canada). Muscle contractions were then elicited by stimulating the distal part of the sciatic nerve via bipolar electrodes, using supramaximal square-wave pulses of 0.2 msec (701 A stimulator; Aurora Scientific). Data acquisition and control of the servomotor were conducted using a Lab-View-based DMC program (version 5.202; Aurora Scientific). Optimal muscle length (Lo) was determined by incrementally stretching the muscle until the maximum isometric twitch force was achieved. The fatigue resistance protocol consisted of 60 tetanic contractions (60 Hz stimulation/500-ms duration) every 3 s for a total of 3 min.

## Automated image analysis

H&E slides were scanned at 20x. For estimating the infiltration of immune cells into skeletal muscles, the Ilastik machine learning software[144] was used to segment the regions with immune infiltration and the total muscle tissue area in each slide, which lead to quantify the ratio of immune infiltration versus the total muscle area. For quanti-fying myofibers with centrally-located nuclei, the StarDist[145] deep learning model was used to segment the nuclei in each slide. Subse-quently, the Ilastik software package was used to generate a stack of 36 features for each slide. We used the feature stacks and the segmented nuclei as an input for the Ilastik object classification workflow and trained a classifier to detect centrally located nuclei. The muscle seg-mentation and immune infiltration mask was used to remove the nuclei located outside the muscle tissue and the nuclei located in the immune-infiltrated muscle regions.

In the graphs that report these quantifications, the center line is the median whereas the lower and upper hinges correspond to the first and third quartiles (the 25th and 75th percentiles). The upper whisker extends from the hinge to the largest value, no further than 1.5 * IQR (inter-quartile range) from the hinge. The lower whisker extends from the hinge to the smallest value, at most 1.5 * IQR from the hinge. The statistical analysis was done with the two-way Mann–Whitney $U$ sta-tistical test, which was run in R with the Wilcox function.

## ELISA assays to quantify the levels of chemokines in the mouse plasma

ELISA assays were done according to manufacturer instructions by using the mouse CXCL1/KC quantikine ELISA kit (R&D, cat. no. MKC00B), the mouse CXCL4/PF4 quantikine ELISA kit (R&D, cat.

no. MCX400), the mouse CXCL5/LIX quantikine ELISA kit (R&D, cat. no. MX000), and the RayBio mouse CXCL7/TCK-1 ELISA Kit (RayBiotech, cat. no. ELM-TCK1-1).

### ELISA assays to quantify the levels of total (active and inactive) versus inactive CXCL7 in injured and uninjured muscles

Total CXCL7 (active and inactive) was quantified from lysates using the RayBio mouse CXCL7/TCK-1 ELISA Kit (RayBiotech, cat. no. ELM-TCK1). Quantification of the uncleaved, inactive form of CXCL7 was performed with a modified version of this kit by using a PPBP polyclonal antibody specific for inactive CXCL7 (Invitrogen, cat. no. PA5-115070, 1:300) as the primary detection antibody and an anti-rabbit HRP-linked secondary antibody (Cell Signaling, cat. no. 7074, 1:3000).

### Neutrophil chemotaxis assays

In vitro chemotaxis assays were done as previously described[73]. Mouse blood was collected by cardiac puncture and red blood cells were lysed for 5 min in lysis buffer (155 mM $NH_4Cl$, 12 mM $NaHCO_3$, 0.1 mM EDTA). The leukocytes were then washed in FACS buffer, blocked with CD16/32 antibody (BioLegend), and stained with fluorophore-conjugated primary antibodies (anti-mouse CD11b and Ly6G). Different recombinant chemokines (rCXCL4, rCXCL5, and rCXCL7; R&D cat. no. 595-P4, 433-MC, and 1091-CK) were added as chemoattractants at 2 μg/mL into the lower chamber of a ChemoTx chemotaxis system (Transwell filter with 5-μm pore size; Neuroprobe). Stained leukocytes were plated in the upper chamber. Both the upper and lower chambers contained RPMI. After 2 h, the content of the lower chamber was collected and mixed with propidium iodide (for assessing cell viability) and Bright Count absolute counting beads (Invitrogen). Samples were then analyzed by FACS to determine the number of migrated $CD11b^+Ly6G^+$ neutrophils.

### Cytokine antibody arrays

TA muscle tissues were homogenized in a bullet blender at 4 °C with 0.5-mm zirconium beads and RayBio Lysis Buffer for antibody arrays (RayBio Lysis Buffer; AA-LYS-10 mL) with protease inhibitors. After homogenization, the lysates were centrifuged for 5 min at 10,000 x g to remove tissue debris and the supernatant was collected and used for probing the cytokine arrays. 10 μL of the supernatant was used for protein quantitation. For each sample, 350 μL (at -1–2 mg/mL) were applied to the Quantibody Mouse Cytokine Antibody Array 640 (RayBiotech, catalog #: QAM-CAA-640), a combination of 16 non-overlapping antibody arrays to quantitatively measure 640 mouse cytokines, and processed by the manufacturer according to the standard procedures listed in the manual for this product. The final concentration of each target cytokine (pg/mL) in each sample was utilized for hierarchical clustering and to generate a heatmap. Specifically, the cytokine heatmap was generated from z-scores of cytokine protein levels, after assigning a base value to each cytokine using 2 × z-score (min non-zero), which was used to replace missing values, i.e., concentrations of 0 pg/mL. Subsequently, a clustering method of UGPMA (unweighted pair group method with arithmetic mean) and similarity measure of correlation were applied, using the Spotfire (v7.5.0, TIBCO) Hierarchical Clustering tool.

### Statistics and reproducibility

Data organization, scientific graphing, and statistical analyses were done with Microsoft Excel (version 14.7.3) and GraphPad Prism (version 8). The unpaired two-tailed Student's $t$ test was used to compare the means of two independent groups to each other. One-way ANOVA with Tukey post hoc testing was used for multiple comparisons of more than two groups of normally distributed data. Two-way ANOVA with post hoc testing (typically Tukey for multiple comparisons between groups, and Sidak for comparisons within a group) was used for multiple comparisons of more than two groups of normally distributed data in presence of two independent variables. The n for each experiment can be found in the figure legends and represents independently generated samples (e.g., TA muscles) sourced from distinct mice. Bar graphs display the mean ± SD. A significant result was defined as $P < 0.05$. Throughout the figures, asterisks and ampersand symbols indicate the significance of $P$ values: $*P < 0.05$, $**P < 0.01$, $***P < 0.001$, $****P < 0.0001$. Ampersand values refer to the comparison of muscles from control versus platelet-depleted or versus Cxcl7ko mice at a given timepoint of regeneration. Representative micrographs are derived from the analysis of multiple muscles.

### Reporting summary

Further information on research design is available in the Nature Portfolio Reporting Summary linked to this article.

## Data availability

All the primary data corresponding to the figures and supplementary figures of this study are available in the Source data File. Source data are provided with this paper.

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

## Acknowledgements

We thank the DSHB for antibodies, and the Light Microscopy facility, the Pathology Core, and the Hartwell Center for Bioinformatics and Biotechnology at St. Jude Children's Research Hospital. M.L. is supported by the National Cancer Institute (R01CA245301) whereas F.D. is supported by the National Institute on Aging (R01AG055532). Research at St. Jude Children's Research Hospital is supported by the ALSAC. The content is solely the responsibility of the authors and does not necessarily represent the official views of the National Institutes of Health.

## Author contributions

F.A.G. did most experiments, with help from A. Stephan and B.A.M.-B.; F.A.G. and A. Shirinifard performed image analysis; Y.-D.W. analyzed cytokine array data; M.L. and F.D. supervised the project, wrote the manuscript, and equally contributed as last and corresponding authors by providing expertise on platelets and skeletal muscle biology, respectively.

## Competing interests

The authors declare no competing interests.
