## [Peer Review File · Nature Communications]

REVIEWER COMMENTS

Reviewer #1 (Remarks to the Author):

This manuscript provides a very nice addition to the emerging story relating neutrophils to skeletal muscle regeneration, filling in a missing piece of that story by demonstrating that platelets are responsible for the early neutrophil accumulation in damaged muscle. Overall the studies and analysis are quite thorough, and conclusions generally supported by the data. There are a few aspects of the manuscript that require clarification, and additional discussion.

Technical concerns:

The data in Fig. 5E are quite important to the conclusions – however, the bars in the figure seem to indicate that the statistically significant differences are between the injured and control (PBS) conditions, not the wild-type and Cxcl7ko mice following injury?

The interpretation of the data from studies using the glycerol injury model are unclear. It appears from Fig. 6A that regeneration is similar between ko and wild-type mice, as measured by muscle mass, but the fiber size distribution is different – is there an increase in the number of muscle fibers in the ko mice or increased fibrosis to account for these findings?

The injury models used in this study will all spontaneously regenerate in wild-type mice. It would be informative for the authors to speculate how their findings may extend to wounds that do not spontaneously heal (e.g., VML injuries).

Reviewer #2 (Remarks to the Author):

In this manuscript by Graca and colleagues, a role for platelets in skeletal muscle regeneration is described. The authors implemented toxin-induced muscle injury models in combination with antibody-mediated platelet depletion and investigated various parameters. Muscle injury led to an increase of (platelet-based) clots inside the damaged muscle, which were absent after platelet depletion. Interestingly, the number of infiltrated neutrophils after injury was decreased when platelets were depleted. Although the potential of muscle regeneration was not declined per se, muscle fibers were thinner 2 weeks after injury in the absence of platelets. A multiplex screening revealed that intramuscular levels of platelet-derived chemokines (notably CXCL1, CXCL2, CXCL4, CXCL5 and CXCL7) were decreased in the absence of platelets. This might explain the reduction of neutrophil infiltration into the tissue. Genetically altered mice with a primary deletion of the CXCL7 gene, but also with

reduced levels of CXCL4 and CXCL5 were subjected to muscle injury (cardiotoxin and glycerol). These mice also showed decreased neutrophil infiltration after injury. In addition, force generation and fiber diameters were reduced 14 days after injury, compared to control mice.

The authors conclude that platelets promote skeletal muscle regeneration by the action of their chemokines on neutrophil recruitment.

The study is interesting, well performed and well presented. The finding that platelet chemokines augment tissue regeneration is novel as far as skeletal muscle is concerned. The authors have convincingly shown that platelets and their chemokines are involved in neutrophil recruitment into injured tissue. However, the steps that lead from neutrophil recruitment to tissue repair are somewhat underexposed. This, and some other conceptual gaps need attention.

As stated above, the link between platelet depletion and neutrophil recruitment is quite solid (also from other studies). But the findings in this study do not unequivocally prove that the absence of neutrophils is responsible for the altered (impaired) muscle healing. There is still a scenario left that platelets are involved in tissue healing for longer periods of time independent of neutrophils, driven by CXCL4, CXCL5 and CXCL7 (that have effects on their own and on many other cell types). For this, the authors might want to consider a depletion or inhibition of neutrophils (or their recruitment) in combination with preserved platelet function, or any well-conceived setup not necessarily involving additional animals.

The proposed involvement of platelet chemokines is very interesting. The authors are correct that the levels of CXCL4 and CXCL7 are >100 times higher than those for CXCL5, CXCL1 and CXCL2. However, CXCL4 is not a classical neutrophil-recruiting chemokine (although some older reports would state that it is) and CXCL7 needs proteolytic processing before it becomes a potent neutrophil attractant. Similar might apply for CXCL4, although less established than for CXCL7. Do the authors have evidence that e.g. CXCL7 is present in its "active" form after muscle injury?

Both CXCL4 and CXCL7 (in its NAP-2 form) are strong inducers of neutrophil extracellular traps (NETs). In addition, CXCL4 is anti-angiogenic and pro-fibrotic. Depending on the cell type, CXCL4 can have pro-mitogenic effects. Have the authors looked at NETs, (neo)vascularisation and fibrosis in their models? It is conceivable that these effects of platelet chemokines are also important in tissue regeneration.

Minor: how many cells are included in the gaussian analyses of fiber thickness?

RESPONSE TO THE REVIEWERS' COMMENTS

Reviewer #1 (Remarks to the Author):

This manuscript provides a very nice addition to the emerging story relating neutrophils to skeletal muscle regeneration, filling in a missing piece of that story by demonstrating that platelets are responsible for the early neutrophil accumulation in damaged muscle. Overall the studies and analysis are quite thorough, and conclusions generally supported by the data. There are a few aspects of the manuscript that require clarification, and additional discussion.

Thank you for the positive evaluation of our work. We have addressed all the points raised, as explained in detail here below.

Technical concerns:

The data in Fig. 5E are quite important to the conclusions – however, the bars in the figure seem to indicate that the statistically significant differences are between the injured and control (PBS) conditions, not the wild-type and Cxcl7ko mice following injury?

Response: We would like to clarify that there is also a significant difference between the wild-type and the Cxcl7ko mice following injury. Specifically, while asterisks (*) are used to indicate the significance between injured and uninjured muscles ($P < 0.01$ and *** $P < 0.001$), ampersands (&) indicate a significant difference in the comparison of wild-type versus Cxcl7ko muscles following injury, which is &&& $P < 0.001$ for Ly6G-stained neutrophils and & $P < 0.05$ for MMP9-stained neutrophils (Fig. 5e).**

The interpretation of the data from studies using the glycerol injury model are unclear. It appears from Fig. 6A that regeneration is similar between ko and wild-type mice, as measured by muscle mass, but the fiber size distribution is different – is there an increase in the number of muscle fibers in the ko mice or increased fibrosis to account for these findings?

Response: Thank you for pointing this out. We have now better discussed these results to convey that although the muscle mass is similar in post-injury Cxcl7ko versus post-injury wild-type mice (Fig. 6a), the histological analyses highlight the presence of debris and empty spaces in-between the myofibers of post-injury muscles from Cxcl7ko mice (Fig. 6d). Therefore, we propose that although there are bigger myofibers in post-injury wild-type muscles compared to post-injury Cxcl7ko muscles (Fig. 6e-f), the muscle mass is similar (Fig. 6a) because there are more debris and interstitial space in-between the myofibers of post-injury Cxcl7ko muscles compared to post-injury wild-type muscles (Fig. 6d).

We have also analyzed the number of myofibers and found no significant difference when comparing post-injury muscles from Cxcl7ko vs. wild-type mice (Fig. 6e). We have also analyzed fat infiltration (via immunostaining for Perilipin-1) but found no significant difference (Supplementary Fig. 7). Therefore, changes in myofiber number and fat infiltration do not seem to play a role in this context.

The injury models used in this study will all spontaneously regenerate in wild-type mice. It would be informative for the authors to speculate how their findings may extend to wounds that do not spontaneously heal (e.g., VML injuries).

Response: We have now better discussed our findings also in the context of volumetric muscle mass loss (VML injuries). Although, we have not performed any experiments with VML models, we hypothesize that the signaling interaction between platelets and neutrophils and/or recombinant platelet-secreted chemokines (e.g. CXCL5/7) may help the regeneration of such injuries by sustaining currently-used interventions such as tissue or stem cell engraftment. Conversely, modulation of platelet-derived chemokine signaling could be utilized to prevent excessive neutrophil infiltration, which occurs in unresolved VML injuries and contributes to the impairment of muscle stem cell function.

Reviewer #2 (Remarks to the Author):

In this manuscript by Graca and colleagues, a role for platelets in skeletal muscle regeneration is described. The authors implemented toxin-induced muscle injury models in combination with antibody-mediated platelet depletion and investigated various parameters. Muscle injury led to an increase of (platelet-based) clots inside the damaged muscle, which were absent after platelet depletion. Interestingly, the number of infiltrated neutrophils after injury was decreased when platelets were depleted. Although the potential of muscle regeneration was not declined per se, muscle fibers were thinner 2 weeks after injury in the absence of platelets. A multiplex screening revealed that intramuscular levels of platelet-derived chemokines (notably CXCL1, CXCL2, CXCL4, CXCL5 and CXCL7) were decreased in the absence of platelets. This might explain the reduction of neutrophil infiltration into the tissue. Genetically altered mice with a primary deletion of the CXCL7 gene, but also with reduced levels of CXCL4 and CXCL5 were subjected to muscle injury (cardiotoxin and glycerol). These mice also showed decreased neutrophil infiltration after injury. In addition, force generation and fiber diameters were reduced 14 days after injury, compared to control mice. The authors conclude that platelets promote skeletal muscle regeneration by the action of their chemokines on neutrophil recruitment. The study is interesting, well performed and well presented. The finding that platelet chemokines augment tissue regeneration is novel as far as skeletal muscle is concerned. The authors have convincingly shown that platelets and their chemokines are involved in neutrophil recruitment into injured tissue. However, the steps that lead from neutrophil recruitment to tissue repair are somewhat underexposed. This, and some other conceptual gaps need attention.

Thank you for the positive evaluation of our work. We have addressed all the specific queries, as indicated below.

As stated above, the link between platelet depletion and neutrophil recruitment is quite solid (also from other studies). But the findings in this study do not unequivocally prove that the absence of neutrophils is responsible for the altered (impaired) muscle healing. There is still a scenario left that platelets are involved in tissue healing for longer periods of time independent of neutrophils, driven by CXCL4, CXCL5 and CXCL7 (that have effects on their own and on many other cell types). For this, the authors might want to consider a depletion or inhibition of neutrophils (or their recruitment) in combination with preserved platelet function, or any well-conceived setup not necessarily involving additional animals.

Response: Thank you for pointing this out. We have now depleted neutrophils by using anti-Gr1 and anti-Ly6G antibodies in mice with preserved platelet function (i.e. without platelet depletion). In agreement with previous studies, we find that neutrophil depletion impairs muscle regeneration: myofiber size and muscle force production are decreased in mice with neutrophil depletion compared to control-treated mice. This data is shown in a new figure (Fig. 7).

The proposed involvement of platelet chemokines is very interesting. The authors are correct that the levels of CXCL4 and CXCL7 are >100 times higher than those for CXCL5, CXCL1 and CXCL2. However, CXCL4 is not a classical neutrophil-recruiting chemokine (although some older reports would state that it is).

Response: We have now examined the neutrophil-chemoattractant capacity of recombinant CXCL4, CXCL5, and CXCL7. As indicated by the reviewer, we find that recombinant CXCL4 has no significant capacity to promote neutrophil migration in vitro, which is instead highly induced by recombinant CXCL5 and CXCL7. This new data is shown in Fig. 4e.

CXCL7 needs proteolytic processing before it becomes a potent neutrophil attractant. Similar might apply for CXCL4, although less established than for CXCL7. Do the authors have evidence that e.g. CXCL7 is present in its "active" form after muscle injury?

Response: We have now analyzed CXCL7 protein levels in injured versus uninjured muscles with different antibodies that recognize total versus inactive CXCL7. Specifically, we have utilized the RayBio CXCL7/TCK-1 ELISA Kit (#ELM-TCK1), which detects total CXCL7 levels, in parallel with the Invitrogen #PA5-115070 antibody, which specifically recognizes the inactive (i.e. unprocessed) CXCL7. By using these tools, we find that total CXCL7 protein levels increase in injured muscles compared to controls whereas inactive (unprocessed) CXCL7 levels decrease upon injury. Overall, these results indicate that CXCL7 activity increases in skeletal muscle upon injury. This new data is shown in Fig. 4f-g.

Both CXCL4 and CXCL7 (in its NAP-2 form) are strong inducers of neutrophil extracellular traps (NETs). In addition, CXCL4 is anti-angiogenic and pro-fibrotic. Depending on the cell type, CXCL4 can have pro-mitogenic effects. Have the authors looked at NETs, (neo)vascularisation and fibrosis in their models? It is conceivable that these effects of platelet chemokines are also important in tissue regeneration.

Response: We have utilized antibodies against citrullinated histone H3 to detect neutrophil extracellular traps (NETs) but did not detect any significant change in the presumed NET immunoreactivity in the muscle sections we examined:

Concerning neovascularization, while CXCL4 is anti-angiogenic there are neutrophil-secreted factors that are pro-angiogenic (such as VEGF). On this basis, the neovascularization observed in regenerating tissues likely depends on the balance of the action of pro-angiogenic and anti-angiogenic factors secreted by both platelets and neutrophils. We have analyzed fat infiltration and neovascularization in the muscles from mice with neutrophil depletion (with normal platelet function) and found that neutrophil depletion leads to impaired regeneration (Fig. 7) that is characterized by a significant decrease in neovascularization (as indicated by the lower abundance of PECAM1-positive blood vessels; Supplementary Fig. 9) which is also observed upon platelet depletion (Supplementary Fig. 6). Upon neutrophil depletion, there is also a non-significant trend towards increased fat infiltration (assessed based on perilipin-1 staining; Supplementary Fig. 9).

Minor: how many cells are included in the gaussian analyses of fiber thickness?

Response: We would like to specify that the gaussian analyses of fiber thickness are based on the inclusion of all myofibers sourced from all muscles. The precise number of myofibers utilized for each plot can be found in the corresponding tab of the Source data file.

REVIEWERS' COMMENTS

Reviewer #1 (Remarks to the Author):

The authors have adequately addressed my earlier concerns.

Reviewer #2 (Remarks to the Author):

I have read the response to reviewer and the revised paper with (renewed) interest. I am very pleased how well the authors have addressed my comments and although I appear to have been the typical reviewer 2, the new findings are a good addition to the general story.

I fully support publication of this excellent work.